



# From soil to sea: Sources and transport of organic carbon traced by tetraether lipids and sediments in the monsoonal Godavari River, India

Frédérique M.S.A. Kirkels[1], Huub M. Zwart[1], Muhammed O. Usman[2,a], Suning Hou[1], Camilo Ponton[3,b],
Liviu Giosan[3], Timothy I. Eglinton[2], Francien Peterse[1]

[1]Department of Earth Sciences, Utrecht University, Utrecht, the Netherlands
[2]Geological Institute, ETH Zürich, Zürich, Switzerland
[3] Geology & Geophysics, Woods Hole Oceanographic Institution, Woods Hole, MA, USA

*[a] present address: Department of Physical & Environmental Sciences, University of Toronto Scarborough, Toronto, Ontario,*
*Canada*
     *[b] present address: Geology Department, Western Washington University, Bellingham, WA, USA*

*Correspondence to*: Francien Peterse (f.peterse@uu.nl)

**Abstract.** Monsoonal rivers play an important role in the land-to-sea transport of soil-derived organic carbon (OC). However, spatial and temporal variation in the concentration, composition, and fate of OC in these rivers remains poorly understood. We
investigate soil-to-sea transport of OC by the Godavari River in India using branched glycerol dialkyl glycerol tetraether (brGDGT) lipids in soils, river suspended particulate matter (SPM), riverbed sediments, and in a marine sediment core from the Bay of Bengal. The abundance and composition of brGDGTs in SPM and sediments in the Godavari River differs between the dry and wet season. In the dry season, 6-methyl brGDGTs dominate SPM and riverbed sediments in the whole basin. Currently, mobilisation and transport of soils from the upper basin is limited due to deficient rainfall and damming. This
promotes aquatic brGDGT production in this part of the basin, which is reflected by a high relative abundance of 6-methyl brGDGTs in both seasons. In the wet season, brGDGT distributions in SPM from the lower basin closely resemble those in soils, mostly from the North and East Tributaries, corresponding to precipitation patterns. The brGDGT composition in SPM and sediments from the delta suggests that soil OC is only effectively transported to the Bay of Bengal in the wet season, when the river plume extends beyond the river mouth. The sediment geochemistry indicates that also the mineral particles exported
by the Godavari River primarily originate from the lower basin, similar to the brGDGTs. River depth profiles of brGDGTs in the downstream Godavari reveal no hydrodynamic sorting effect in either season, indicating that brGDGTs are not associated with certain minerals. The similarity of brGDGT distributions in bulk and fine-grained sediments (≤63μm) further confirms the absence of selective transport mechanisms. Nevertheless, the composition of brGDGTs in a Holocene, marine sediment core near the river mouth appears substantially different from that in the modern Godavari basin, suggesting that terrestrial-
derived brGDGTs are rapidly lost upon discharge into the Bay of Bengal and/or overprinted by marine in situ production. The





change in brGDGT distributions at the river-sea transition implies that this zone is key in the effective transfer of soil OC, as well as for the interpretation of paleorecords based on brGDGTs in coastal marine sediment archives.

## 1 Introduction

Fluvial land-to-sea transport of organic carbon (OC) plays a key role in the global carbon cycle (Aufdenkampe et al., 2011; Bianchi et al., 2011; Galy et al., 2015; Ward et al., 2017). The burial of river-exported OC in marine sediments results in the long-term sequestration of photosynthetically-fixed $CO_2$ from the atmosphere (Hedges an Keil, 1995; Hedges et al., 1997; Leithold et al., 2016). On land, soils are the most important OC pool (1500-2400 Gt), as its size exceeds the atmospheric and biotic inventories (>2.2 and 2.3 times, respectively) (Friedlingstein et al., 2020). Soil organic carbon (SOC) is continuously mobilised by erosion processes and transferred into rivers, where it forms a key component of fluvial OC (Holtvoeth et al., 2005; Tao et al., 2016). Rivers were long considered 'pipelines' or 'passive channels' where SOC remains unchanged during transport downstream and represents an integration of the whole river basin (Cole et al., 2007). However, it appears that only a fraction (30-50%) of the SOC that enters a river ultimately reaches the ocean (Cole et al., 2007; Battin et al., 2009; Aufdenkampe et al., 2011; Bianchi et al., 2011; Ward et al., 2017), and that its composition can be highly altered during transport (Hedges, 1992; Hedges et al., 1997).

The pathway of SOC through a river basin, i.e., where, when, and to what extent it is mobilised, processed enroute, or transferred downstream and exported to the ocean, is determined by a combination of physical and biogeochemical processes along the soil-river-ocean continuum (Cole et al., 2007; Battin et al., 2009; Aufdenkampe et al., 2011; Butman and Raymond, 2011; Regnier et al., 2013; Ward et al., 2017). For example, the stability of SOC during transport, and thus its potential for final preservation in marine sediments can be influenced by the mode of transport: in free (uncomplexed) form, or associated with mineral surfaces (e.g., organo-mineral complexes) due to differences in transport efficiency, sinking velocity, and physical protection against microbial and oxidative attack (Mayer, 1994; Keil et al., 1997; Blattmann et al., 2019; Hemingway et al., 2019). Changes in the sourcing of SOC from specific parts of a basin corresponding to temporal variations in rainfall patterns are known to influence OC that is finally exported to the ocean (e.g., Galy et al., 2008; Hemingway et al., 2017; Menges et al., 2020). In addition, hydrodynamic particle sorting can cause stratification of sediment loads with depth and can thus affect bulk OC transport (Galy et al., 2008; Bouchez et al., 2011, 2014), although influences on specific OC components may vary (Freymond et al., 2018a). Notably, terrigenous OC that is moved through a river basin can be a composite of soil- (pedogenic) and rock-derived (petrogenic) OC, with in situ aquatic productivity as additional OC source (e.g., Eglinton et al., 2021). Disentangling the sources of river-transported OC is difficult using bulk measurements, and as a result, the timescales and mechanisms (quantitative and qualitative) of transport, as well as the composition of the OC that is finally discharged to the ocean, remain elusive for many river systems. Although lipid biomarkers represent only a small part of the total OC, their source- or environmental-specific signature renders them as promising tracers in river systems (e.g., Eglinton et al., 2021).



Over the past decade, bacterial membrane lipids, so-called branched glycerol dialkyl glycerol tetraethers (brGDGTs), have been used to trace soil organic matter inputs into rivers and continental margins (e.g., Hopmans et al., 2004; Weijers et al., 2009a; Kim et al., 2012; Zell et al., 2013a,b; Kirkels et al., 2020a; Märki et al., 2020). BrGDGTs occur globally in soils and

peats, and can vary in the number (4-6) and position of methyl branches (5/5' or 6/6', defined as 5-methyl and 6-methyl) attached to linear $C_{28}$ alkyl chains, as well as the presence of 0-2 cyclopentane moieties (Weijers et al., 2007a; De Jonge et al., 2013, 2014a; see figures therein for the molecular structures). Although their exact producers remain enigmatic, the stereochemistry of the glycerol moieties of brGDGTs points toward a bacterial source (Weijers et al., 2006), likely Acidobacteria (Weijers et al., 2009b; Sinninghe Damsté et al., 2011). Iso-diabolic acids, which are considered building blocks

of brGDGTs, have been detected in Acidobacteria subdivsions 1, 3, 4 and 6 (Sinninghe Damsté et al., 2014, 2018). Whereas most subdivisions produce iso-diabolic acid with a methylation on the 6/6' position, 5-methyl iso-diabolic acid appears so far to be exclusively produced by Acidobacteria from subdivision 4 (Sinninghe Damsté et al., 2014, 2018). The identification of Acidobacteria that produce intact brGDGTs was so far limited to two cultured species of subdivision 1 that synthesise brGDGT Ia (Sinninghe Damsté et al., 2011), but recently a member of subdivision 3 was reported to produce a whole suite of 5-

methylated and cyclic brGDGTs as their main lipids (Chen et al., 2022).

Studies on the occurrence and distributions of brGDGTs in a large set of global surface soils have revealed empirical relations between the degree of methylation of 5-methyl brGDGTs (MBT'$_{5me}$) and mean annual air temperature (MAAT), and between the degree of cyclisation (CBT'), as well as the relative abundance of 6-methyl brGDGTs and soil pH (Weijers et al., 2007a; De Jonge et al., 2014a; Naafs et al., 2017). These correlations can subsequently be used to determine MAAT and soil pH based

on brGDGT signals in environmental samples. For example, downcore variations in brGDGT distributions in Congo River fan sediments have yielded a record of deglacial warming for tropical Africa (Weijers et al., 2007b). Furthermore, the presence of brGDGTs in coastal marine sediments relative to that of crenarchaeol, an isoprenoid GDGT produced by marine archaea (Sinninghe Damsté et al., 2002), has been used as a proxy to determine the relative contribution of terrestrial OC into a marine environment, and is quantified in the branched and isoprenoid tetraether (BIT) index (Hopmans et al., 2004). These proxies

are based on the assumption that brGDGTs preserved in continental margin sediments have a soil origin and represent a basin-integrated signal. Indeed, the composition of brGDGTs in rivers has been shown to resemble those in soils in upstream catchments (e.g., Kirkels et al., 2020; Märki et al., 2020). However, brGDGTs have also been found to be produced in rivers (e.g., Yang et al., 2013; Zell et al., 2013a,b; De Jonge et al., 2014b, 2015a), which complicates their use as soil-specific tracers. Zell et al. (2013a,b) reported a mismatch between brGDGT compositions in soils and river suspended particulate matter (SPM)

from the Lower Amazon River in the dry season, and attributed this difference to in situ aquatic brGDGT production.

Next to changes in the origin i.e., soil or aquatic, hydroclimate variability may also influence the provenance i.e., source location within a basin of brGDGTs in a river basin. For example, brGDGT signals of SPM from the Congo River matched with different areas of the river basin in response to seasonal changes in rainfall distribution (Hemingway et al., 2017). Further progress on untangling brGDGT sources has been made since De Jonge et al. (2014b) discovered a potential link between the

occurrence of 6-methyl brGDGTs and aquatic production in rivers based on predominance of these isomers in SPM of the





Yenisei River compared to local soils. The abundance of 6-methyl brGDGTs relative to 5-methyl isomers can be quantified in the isomer ratio (IR), where higher IR values appear to be indicative of more aquatic production. Following this principle, Guo et al. (2020) determined that at least 65% of the brGDGTs in sediments from the Carminowe Creek in southwest England were produced in situ. Similarly, higher IR values in SPM than in catchment soils in the Madre de Dios River, an upper tributary of

the Amazon River, could be linked with enhanced aquatic production in the dry season, when soil input into the river is limited (Kirkels et al., 2020a).

Upon discharge into the marine realm, the soil-derived brGDGT signal may be further altered by coastal marine brGDGT production. Such a marine contribution to the pool of brGDGTs can be identified using the weighted number of cyclopentane moieties in tetramethylated brGDGTs ($\#rings_{tetra}$), where values >0.7 indicate a purely marine origin (Sinninghe Damsté,

2016). As such, $\#rings_{tetra}$ has been used to reconstruct changes in the origin of brGDGTs over the Holocene in Baltic Sea sediments (Warden et al., 2018), and to correct for marine 'overprint' on the terrestrial climate signal of soil-derived brGDGTs in Pliocene sediments from the North Sea basin (Dearing Crampton-Flood et al., 2018).

In this study, we analyse a large set of soils, river SPM, riverbed and estuarine sediments collected from the main stem and major tributaries of the Godavari River in Peninsular India in a pre-monsoon (dry) and monsoon (wet) season to explore the

evolution of brGDGT signatures during transport along the land-river-sea continuum. The distinct wet and dry seasons in this river system allows us to investigate seasonal changes in the origin of brGDGTs, i.e., soil-derived or aquatic produced. In addition, different bedrock types in the upper and lower parts of the Godavari basin enable us to determine the provenance of the mineral fraction, by looking at changes in the elemental composition of the riverbed sediments. Consequently, the mode of transport (free or mineral-associated) of the brGDGTs can be assessed by direct comparison with the mineral elemental

composition of the same soils and sediments (bulk and fine, i.e., ≤0.63 μm) they are extracted from. Possible influences by hydrodynamic sorting and/or selective transport of certain brGDGTs are evaluated based on their occurrence and distribution along multiple depth-profiles in the main stem Godavari and the delta. Finally, downcore variations in brGDGT distributions in Holocene sediments from the Bay of Bengal give insight into the transfer efficiency of fluvially-discharged brGDGTs to the marine sedimentary archive, and thus the reliability of brGDGT-based paleorecords obtained from such archives.

**2 Study area**

The Godavari River is the largest monsoon-fed river of Peninsular India. The Godavari has its source in the Western Ghats mountain range and flows eastward across central India before reaching the Bay of Bengal (Fig. 1). The Godavari has 5 major subbasins with a distinct hydrology and geology: the Upper (~37% of the total basin area), Middle (6%) and Lower (2%) Godavari cover the main stem of the river, whereas the North (35%) and East Tributaries (20%) comprise contributions by the

Wainganga, Penganga, Wardha and Pranhita Rivers, and the Indravati and Sabari Rivers, respectively (Babar and Kaplay, 2018) (Fig. 1b,c). We here refer to the Upper Godavari and the North Tributary headwaters (i.e., Wardha and Penganga Rivers)



that drain the Deccan plateau as the upper basin, and to all other North Tributaries, East Tributaries, and Middle and Lower Godavari as the lower basin.

The Godavari River has a catchment area of $3.1 \times 10^5$ km$^2$, a length of 1465 km, and exports ~2.8 Mt of OC and 170 Mt of

sediment to the Bay of Bengal annually (Biksham and Subramanian, 1988a,b; Gupta et al., 1997; Babar and Kaplay, 2018). Currently, a dam with a Reservoir Lake built in the mid-19th century at Rajahmundry controls the flow to the tidally-influenced delta. In the delta, the river splits into three branches, of which the northern (Gautami) branch carries the majority (67%) of the discharge and sediment load via an estuary to the Bay of Bengal (Rao et al., 2015). Smaller dams that control the river flow year-round are abundant in the upper basin (Pradhan et al., 2014).


**Figure 1 – (a)** Location of the Godavari River basin in peninsular India, **(b)** Sampling sites along the river basin, with the different subbasins (grey) and upper and lower basin (red) shown. **(c)** Zoom for the Godavari delta and the Reservoir Dam at Rajahmundry. **(d)** Precipitation across the Godavari basin based on the APHRODITE data set (Asian Precipitation – Highly Resolved Observational Data Integration Towards Evaluation of Water Resources, V1101 Monsoon Asia) of mean annual precipitation for

30 years at a 0.25° grid resolution (Yatagai et al., 2009). **(e)** Geological context of the Godavari basin.



The hydroclimate of the Godavari River basin is characterised by a distinct dry period (October-May) and a monsoon (wet) season (June-September). The Southwest Monsoon, with its main moisture source in the Indian Ocean and Arabian Sea, brings 75-85% of the annual rainfall to the Godavari basin in the wet season (Balakrishna and Probst, 2005; Kirkels et al., 2020b).

Approximately 94% of the annual discharge and ~98% of the sediment load is exported during the monsoon (wet) season (Rao et al., 2015). The Upper Godavari subbasin lies in the rain shadow of the Western Ghats mountain range and receives limited precipitation (~430 mm year$^{-1}$). The semi-arid climate of the upper basin gradually transitions to (sub-)humid conditions with increasingly more precipitation (~2300 mm year$^{-1}$) in the lower reaches (Gunnell, 1997; Babar and Kaplay, 2018) (Fig. 1d). Rainfall data from the year of sampling can be found in Kirkels et al. (2020b), and discharge data in Appendix 1.The mean

annual air temperature in the basin averages around 27°C (Dearing Crampton-Flood et al., 2019a).

The headwaters of the Upper Godavari (source to Nanded) and of the North Tributaries (Wardha and Penganga Rivers) drain the Deccan plateau (Fig. 1e). This plateau consists of flood basalts of ~200 to 2000 m thickness formed by volcanic activity across the Cretaceous-Tertiary boundary (65-66 Ma) that have weathered into thick, highly erodible clay loam layers (Biksham and Subramanian, 1988a,c; Keller et al., 2008; Meert et al., 2010), rich in iron oxides and clay minerals (predominantly

smectite) (Das and Krishnaswami, 2007; Babechuk et al., 2014; Usman et al., 2018). The lower stretches of the Upper Godavari flow through (felsic) granite and gneiss complexes. The Middle Godavari and the Pranhita and Wainganga Rivers in the North subbasin drain Archean and Proterozoic (Precambrian) aged sedimentary and metamorphic rocks, Permian-Mesozoic Gondwana sediments which are mildly metamorphosed, mature shales, and (feldspathic) sandstones of Neogene age (Meert et al., 2010; Amarasinghe et al., 2015). This region is further characterised by active open-pit mining of near-surface coal deposits

(Singh et al., 2012; Pradhan et al., 2014).

The East Tributaries drain the Eastern Ghats mountains consisting of erosion-resistant, high-grade metamorphic rocks of Precambrian age, including khondalites (Si-Mg rich), charnockites (quartz-feldspar), gneisses, and feldspar rich granulites (Meert et al., 2010; Amarasinghe et al., 2015; Manikyamba et al., 2015). The Lower Godavari River traverses thick layers of alluvium formed by Quaternary river deposits, underlain by Gondwana sandstones (Manikyamba et al., 2015). Source rock

weathering in the lower reaches results in quartzo-feldspathic minerals and minor amounts of kaolinite (Usman et al., 2018). Soils range from 10-40 cm deep, clay-dominated, alkaline vertisols interspersed with shallow leptosols in the Upper Godavari subbasin, to deeper-developed, weathered, acidic lixisols and nitisols ('red soils') towards the east, and sandy fluvisols and arenosols towards the delta (Biksham and Subramanian, 1988b; WRB FAO, 2014; Giosan et al., 2017).

## 3 Materials and methods

### 3.1 River basin sampling

Surface soils (0-10 cm) were collected with a shovel from undisturbed sites near the Godavari River and its main tributaries during the dry season in February/March 2015 (n=46), combining 3-5 spatial replicates (Fig. 1b,c). These topsoils represent





material that may be eroded and transported into the river by next precipitation events. Suspended Particulate Matter (SPM) was sampled during both dry (February/March 2015, n=50) and wet seasons (July/August 2015, n=56). At each sampling location, 20-80 L of surface water was collected at mid-channel position by immersion of a bucket from a bridge or boat or otherwise from 2-3 m out of the riverbank. River depth profiles (2-3 depths, 1-3 sites across river) were sampled in the Godavari delta (main branch, dry and wet season) and in the main stem in the Middle Godavari (wet season) (Fig. 1b,c; site 10 and 28). At these sites, riverwater was collected with equal increments to the riverbed as monitored by a mounted sonar instrument (Humminbird PiranhaMAX 153), using a custom-built depth sampler (after Lupker et al., 2011). After collection, the riverwater was transferred into pre-rinsed containers, and filtered over pre-combusted (450°C, 6h) and pre-weighted 0.7 μm glass fibre filters (GFF) (Whatman, United Kingdom) using pressurised steel filtration units (after Galy et al., 2007) to obtain the SPM. Riverbed sediments were collected at mid-channel position during both dry (n=37) and wet (n=37) seasons, using an Eijkelkamp sediment grabber (Van Veen grab 04.30.01, The Netherlands) or a shovel when the water level was low. At selected sites, soils (n=10) and wet season riverbed sediments (n=25) were sieved in the field to isolate the fine fraction ≤63 μm.

All SPM filters were kept at 4°C and soils and sediments at ambient temperature after collection and subsequent transport to the laboratory at Utrecht University, where all materials were stored frozen (-20°C) and then freeze-dried. SPM filters were weighted to determine sediment loads. Bulk soils and riverbed sediments were homogenised by manual removal of stones and plant debris, and then ground to a fine powder using a stainless-steel/agate mortar or ball mill (OC and biomarker analysis) or a Herzog mill (elemental analysis). Sampling procedures for the marine sediment core NGHP-01-16A (16.59331°N, 82.68345°E, 1268 m water depth, n=46) in the Bay of Bengal are described in Ponton et al. (2012) and Usman et al. (2018).

### 3.2 pH measurements

Riverwater pH was measured in the field with an HQ40d multi-parameter meter (Hach, USA), fitted with a pH probe (IntelliCAL PHC101). The pH of bulk soils and riverbed sediments was measured upon return to the laboratory, using a SympHony SB70D with pH probe (VWR, USA) and distilled water to create a 1:5 (v/v) solid to water mixture after shaking for 30 min at 200 RPM and settling overnight (~18h at 4°C).

### 3.3 Total organic carbon analysis

The Total Organic Carbon (TOC) content of bulk soil, bulk riverbed sediments, and SPM was analysed with a Flash 2000 Organic Element Analyser equipped with a MAS 200 Autosampler (Thermo Scientific, Italy) at NIOZ (Texel, The Netherlands). Prior to analysis, powdered riverbed and soil samples were decalcified by adding 1M HCL, mixed and subsequently rinsed 2 times with distilled water and dried at 60°C. For SPM, pieces of GFF filters were randomly selected and placed in pre-combusted (450°C, 6h) Ag capsules. These capsules were put in a desiccator at 70°C with 37% HCl for 72h (decalcification) and subsequently dried for minimal 120h with NaOH (neutralisation) (Komada et al.2008; van der Voort et al., 2016). Fine fraction (≤63 μm) riverbed sediments and soils were analysed with a NC2500 Elemental Analyser



(ThermoQuest, Germany) at VU University (Amsterdam, The Netherlands). These fine fractions were placed in pre-combusted Ag cups and acidified in situ by adding 1M HCl and subsequently dried overnight at 60°C (Vonk et al., 2008, 2010). TOC results were normalised to internal standards (Acetanillide and Benzoic acid at NIOZ and USGS40, USGS41 and IAEA601 at VU University), with an analytical reproducibility better than 0.07% based on replicate analysis of standards and samples.

**3.4 Soil and sediment elemental composition**

Major and trace elements were measured by Inductively Coupled Plasma-Optical Emission Spectrometry (ICP-OES) using a Spectro Arcos (Ametek, Germany) at Utrecht University for soils and riverbed sediments (bulk and fine fractions). Approximately 100-125 mg of freeze-dried and powdered soil or sediment was digested in a 2.5 mL acid mixture ($HClO_4$:$HNO_3$; 3:2 (v/v)) with 2.5 mL 48% HF, and heated at 90°C overnight. Next, the acid mixture was evaporated at 140°C, and the residue was dissolved overnight in 1M $HNO_3$ at 90°C for elemental analysis. Replicate analyses of selected
samples gave a precision of ±3% for the major elements and ±10% for Titanium.

**3.5 GDGT extraction and analysis**

All bulk and fine fraction soils and riverbed sediments were analysed for GDGTs. For SPM, a selection was made for the dry (n=20) and wet seasons (n=49). The freeze-dried and powdered soils (∼2–20 g), riverbed sediments (∼1–18 g) and SPM (∼0.04–5 g on filter pieces) were extracted with dichloromethane (DCM): methanol (MeOH) (9:1, v/v) using an Accelerated
Solvent Extractor (ASE 350, Dionex) at 100 °C and 7.6 × 106 Pa. The total lipid extracts (TLEs) were dried under a gentle stream of $N_2$. TLEs of wet season SPM and bulk soils were saponified with KOH in MeOH (0.5 M, 2h at 70°C). Separation was enhanced by adding distilled water with NaCl, after which the neutral fraction was back-extracted with Hexane (3x 10 mL). The remaining mixture was acidified to pH ∼2 by adding 1.5 M HCl (dissolved in MeOH) and back-extracted (3x 10 mL) with Hexane:DCM (4:1, v/v) to isolate the acid fraction containing fatty acids for further use by Usman et al. (2018). For
our study, the neutral fraction was passed over a $Na_2SO_4$ column to remove water and further separated over an activated $Al_2O_3$ column into an apolar and polar fraction using Hexane and DCM:MeOH (1:1, v/v) as eluents, respectively. TLEs of dry season SPM, riverbed sediments (bulk and fine fraction ≤63 μm) and fine fraction soils were directly fractionated into an apolar, neutral and polar fraction by passing over an activated $Al_2O_3$ column using Hexane, Hexane:DCM (1:1, v/v) and DCM:MeOH (1:1, v/v) as eluents, respectively.

All polar fractions, containing the GDGTs were spiked with a $C_{46}$ GTGT as internal standard (Huguet et al., 2006), dried, re-dissolved in hexane:isopropanol (99:1, v/v), and passed over a 0.45 μm polytetrafluoroethylene (PTFE) filter prior to analysis. GDGTs were measured using Ultra High Performance Liquid Chromatography-Atmospheric Pressure Chemical Ionisation Mass Spectrometry (UHPLC-APCI-MS) with an Agilent 1260 Infinity coupled to an Agilent 6130 quadrupole mass detector (Agilent Technologies, USA) at Utrecht University with settings according to Hopmans et al. (2016).

In brief, GDGTs were separated over two silica Waters Acquity UPLC BEH Hilic columns (150 × 2.1 mm; 1.7μm; Waters corp., USA) maintained at 30 °C, preceded by a guard column packed with the same material. The compounds eluted



isocratically with 82% A and 18% B for 25 min at 0.2 mL min$^{-1}$, followed by a linear gradient to 70% A and 30% B for 25 min and then to 100% B in 30 min, where A = hexane and B = hexane:isopropanol (9:1, v/v). Sample injection volume was 10 µL. ACPI settings were as follows: gas temperature 200 °C, vaporiser temperature 400 °C, drying gas (N$_2$) flow 6 L min$^{-1}$, capillary

voltage 3500 V, nebuliser pressure 60 psi, corona current 5.0 µA. Detection was achieved in selected ion monitoring mode (SIM), using m/z 744 for the standard, m/z 1292 for crenarchaeol and m/z 1050, 1048, 1046, 1036, 1034, 1032, 1022, 1020 and 1018 for brGDGTs.

Core samples (NGHP-01-16A) were extracted by Usman et al. (2018), and the polar fractions were analysed for only brGDGTs at NIOZ (The Netherlands) using the same UHPLC method and settings. The same core was extracted previously by Ponton

et al. (2012) at a lower resolution and analysed for both isoprenoid and branched GDGTs at ETH Zürich (Switzerland) using a Grace Prevail Cyano column (150 x 2.1 mm; 3µm) and settings according to Schouten et al. (2007), that do not separate 5- and 6-methyl brGDGTs. Agilent Chemstation software (B.04.03) was used to integrate peak areas in the mass chromatograms of the protonated molecule ([M+H]$^+$).

### 3.6 Proxy calculations

BrGDGT indices were calculated using the fractional abundances of specific brGDGTs. Roman numerals in all formulas refer to the molecular structures of brGDGTs shown in De Jonge et al. (2014a) and of isoprenoid GDGTs in Kim et al. (2010). Spatial and temporal variations in brGDGT signals were assessed using the degree of cyclisation of branched tetraethers, calculated according to De Jonge et al. (2014a):

$$CBT' = {}^{10}log((Ic + IIa' + IIb' + IIc' + IIIa' + IIIb' + IIIc') / (Ia + IIa + IIIa)) \qquad Eq. (1)$$

As well as the degree of methylation of 5-methyl brGDGTs:

$$MBT'_{5me} = (Ia + Ib + Ic) / (Ia + Ib + Ic + IIa + IIb + IIc + IIIa) \qquad Eq. (2)$$


Potential in situ production of brGDGTs within the river was assessed using the isomer ratio (IR) of penta- and hexamethylated brGDGTs (Dang et al., 2016, modified from De Jonge et al., 2014b):

$$IR = (IIa' + IIb' + IIc' + IIIa' + IIIb' + IIIc') / (IIa + IIb + IIc + IIIa + IIIb + IIIc + IIa' + IIb' + IIc' + IIIa' + IIIb' + IIIc')$$

$$Eq. (3)$$

To identify contributions of marine in situ production of brGDGTs, the weighted average number of cyclopentane moieties of the tetramethylated brGDGTs was calculated following Sinninghe Damsté (2016):





#rings$_{tetra}$ = (Ib + 2 × Ic) / (Ia + Ib + Ic)                                                    Eq. (4)

As well as the ratio of acyclic hexa- to pentamethylated brGDGTs, including 6-methyl brGDGTs (Xiao et al., 2016, 2020):

Σ IIIa / Σ IIa = (IIIa + IIIa') / (IIa + IIa')                                                              Eq. (5)

The branched and isoprenoid tetraether (BIT) index was calculated following Hopmans et al. (2004), adjusted to include 6-methyl brGDGTs:

BIT index = (Ia + IIa + IIa' + IIIa + IIIa') / (Ia + IIa + IIa' + IIIa + IIIa' + Crenarchaeol)          Eq. (6)

Based on brGDGTs in the marine sediment core spanning the Holocene period, mean annual air temperatures (MAAT) for the Godavari basin were reconstructed using the MBT'$_{5me}$ index and the BayMBT$_0$ model (MatLab R2020a, v.9.8.0.1323502), following Dearing Crampton-Flood et al. (2020). A prior mean of 27.2°C, which is the average measured MAAT for the Godavari basin extracted from the 0.5° gridded CRU TS v. 3.24.01 dataset (Dearing Crampton-Flood et al., 2020), and a prior
standard deviation of 10°C were chosen as model input.

Sea surface temperatures (SSTs) were estimated based on isoGDGTs extracted from the NGHP-01-16A core by Ponton et al. (2012) and the relation between TEX$_{86}^H$ and SST derived by Kim et al., 2010:

TEX$_{86}^H$ = log((GDGT-2 + GDGT-3 + Crenarchaeol') / (GDGT-1 + GDGT-2 + GDGT-3 + Crenarchaeol'))

Eq. (7)

SST = 68.4 × TEX$_{86}^H$ + 38.6                                                                           Eq. (8)

**3.7 Statistical analysis**

The significance of differences in brGDGT concentrations, distributions, and proxy values for each sample type, season, or location in the basin was evaluated with (Welch's) ANOVA and t-tests in R software package for statistical computing (RStudio, v. 1.2.5033; R4.0.4) and SPSS (IBM, v. 26.0.0.0). The level of significance was $p \leq 0.05$. Results are reported as mean ± standard error (SE). Spatial patterns were further investigated with ArcGIS 10.6.1 software (ESRI, USA). Principal
Component Analysis (PCA) was performed in the R-package FactoMineR (Lê et al., 2008) using the relative abundances of the most abundant brGDGTs (i.e., Ia, Ib, Ic, IIa, IIa', IIb', IIIa and IIIa'). Linear correlations were assessed by Pearson correlation coefficients.



## 4 Results

### 4.1 Physical and chemical properties of Godavari River water, SPM, sediments, and soils

#### 4.1.1 Suspended sediment load

All geochemical data can be found in Kirkels et al. (2021a). Suspended sediment loads in surface waters of the Godavari ranged from 1 to 135 mg L-1 in the dry season (14±4 mg L$^{-1}$ (mean ± SE), n=41) and from 2 to 1435 mg L-1 in the wet season (149±37 mg L$^{-1}$, n=40) (Fig. 2a). Average suspended loads were significantly higher in the wet season (p≤0.001), with concentrations up to 75 to 170 times higher at certain sites compared to the dry season, especially in the Pranhita River (North

Tributaries), the Middle, and Lower Godavari. In contrast, the Upper Godavari and those North Tributaries that drain the Deccan plateau, had low suspended loads (mostly <50 mg L$^{-1}$). In the dry season, spatial variability in sediment load was minor. The depth profiles taken in the main stem Godavari and the delta (site 28 and 10, respectively; Fig 1b,c) show that suspended sediment concentrations did not substantially vary with depth in both the dry and wet season (Fig. 3a). A slight increase in suspended load above the riverbed was only observed at the site closest to the eroding riverbank in the Godavari

delta (Fig. 3a).

### 4.1.2 Total organic carbon

The organic carbon content (%OC) in soils was between 0.2 and 1.9% (0.8±0.1, n=46) for the bulk and between 0.5 and 1.0% (0.8±0.1%, n=10) for the fine fractions (Fig. 2b). The %OC was generally higher in the fine fraction than in bulk soils from the same site (0.82±0.05% vs 0.69±0.07%, p=0.06, n=10). There was no spatial trend in %OC in soils across the basin.

The %OC in surface SPM collected during the dry season varied between 1.1 and 32.4% (11.4±1.1%, n=39) and was significantly higher than that in wet season SPM (1.4-20.1%, 5.2±0.7, n=40, p≤0.001). In both seasons, the %OC of surface SPM from the upper basin was significantly higher than that from the lower basin (dry: 15.5±2.2%, n=13 vs 9.4±1.0%, n=26, p≤0.01; wet: 8.4±1.5%, n=14 vs 3.5±0.5%, n=26, p≤0.01). In the dry season, the %OC of surface SPM decreased downstream, changing from 15.2±2.3% (n=11) in the Upper Godavari, 12.4±2.1% (n=10) in the North and 7.6±1.1% (n=6) in the East

Tributaries to 13.3±2.8% (n=4) and 7.0±1.9% (n=8) in the Middle and Lower Godavari, respectively. In the wet season, the %OC of SPM showed a stronger decrease downstream (p≤0.05), and changed from 9.2±1.6% (n=12) in the Upper Godavari, 2.5±0.2% (n=9) in the North and 6.6±3.1% (n=4) in the East Tributaries, to 4.0±0.8% (n=5) in the Middle and 2.8±0.2% (n=10) in the Lower Godavari. The %OC in SPM showed minor variation (<1% and mostly <0.5%) along the depth profiles collected in both the dry and wet season (Fig. 3b).

Bulk riverbed sediments had on average a lower %OC in the dry season (0.4±0.1%, 0.04-19.5%, n=37) than in the wet season (0.6±0.1%, 0.03-3.1%, n=37). The riverbed sediments showed no spatial trend in %OC across the Godavari basin in either season. Fine fractions collected in the wet season had a significantly higher %OC than the bulk riverbed sediments at the same sites (fine: 1.2±0.1%, bulk: 0.7±0.2%, n=25, p≤0.01).




**Figure 2 – Box-and-whisker plots of the organic, geochemical and brGDGT data for the Godavari basin and marine sediments in the Bay of Bengal. (a) Suspended sediment load, (b) Total Organic Carbon (TOC), (c) measured pH, and (d) brGDGT concentrations in the upper and lower basin and in Holocene marine sediments. (e) Ti, (f) Fe and (g) K amounts in the Deccan basalt and felsic bedrock regions, and * in shallow (0-48 cm below sea floor, ~300 yr.; Kalesha et al., 1980) and deep marine sediments (0-300 m**

**(NGHP-1-3B) and 0-184 m bsf (NGHP-1-5C), no age model used; Mazumdar et al., 2015). (h) Relative abundance of brGDGT Ia, (i) Relative abundance of brGDGT IIa', (j) BIT index, and (k) Isomer Ratio (IR) values in the upper and lower basin and in Holocene marine sediments. The box represents the first (Q1) and third (Q3) quartiles, and the line in the box represents the median value, the whiskers extent to 1.5×(Q3-Q1) values and outliers are shown as points. Solid lines represent bulk data, dashed lines the fine fractions (≤63 µm). The pH represents values measured in soil and sediment extracts and in surface water for SPM.**



### 4.1.3 Measured pH

The pH measured in soils in the Godavari basin ranged from 6.0 to 9.1 (n=46), with generally higher pH (mostly >8.0) in the soils from the upper basin and lower pH (mostly <8.0) in the lower basin (Fig 2c). The in situ measured pH of surface waters and the pH of riverbed sediments was on average slightly higher in the dry season (water: 8.4±0.1, n=46; sediment: 8.6±0.1, n=37) than in the wet season (water: 8.2±0.0, n=47, p≤0.03; sediment: 8.3±0.1, n=37, p≤0.09). In both seasons, riverwater pH was relatively constant with river depth (Fig. 3c).

### 4.1.4 Soil and sediment elemental composition

The elemental composition of soils and sediments is different between those parts of the Upper Godavari and North Tributary headwaters that drain the Deccan basalts and the lower reaches of the Godavari basin that drain felsic metamorphic and sedimentary rocks (Fig. 2e-g). In particular, Ti and Fe concentrations in bulk soils from the Deccan basalts were significantly higher than those formed on felsic rocks in the lower part of the basin (Ti: 0.34±0.03, n=13 vs 0.21±0.02 mmol, n=34, p≤0.001; Fe: 1.50±0.08 vs 0.97±0.06 mmol, p≤0.001). In contrast, K concentrations were significantly lower in Deccan soils than those formed on felsic rocks (0.16±0.01 vs 0.41±0.03 mmol, p≤0.001). Riverbed sediments showed the same spatial division in these elements, although less pronounced for Ti and Fe in the wet season. Comparison of bulk and fine fraction soils revealed no differences (p≥0.05), whereas fine fraction sediments in the wet season contained more Ti and Fe and less K compared to the bulk sediments (p≤0.01).

### 4.2 Concentrations and distribution of GDGTs in soils, SPM, riverbed sediments of the Godavari River, and downcore Bay of Bengal sediments

### 4.2.1 GDGTs in catchment soils

The soils analysed in this study (n=46) were previously included in a global dataset to develop a Bayesian calibration model (Dearing Crampton-Flood et al., 2019a, 2020), but brGDGT concentrations and distributions have not yet been described in detail. In short, brGDGTs were detected in all Godavari soils and ranged in concentration from 0.4 to 29.5 µg g-1 OC (5.9±0.8 µg g$^{-1}$ OC), without a clear spatial trend across the basin (Fig. 2d). BrGDGT concentrations in fine fractions averaged 6.5±1.7 µg g$^{-1}$ OC (0.8-18.7, n=10), and are comparable to that in the bulk soil at the same site. Bulk soils were dominated by tetramethylated brGDGTs (relative abundance 56-97%, average 74±1%), followed by penta- (3-41%, average 24±1%) and hexamethylated (0-6%, 2±0%) compounds (Fig. 4a). BrGDGT Ia was the most abundant compound (18-92%, 50±2%) contributing to >50% of the brGDGTs in soils along the Pranhita River (North Tributary), East Tributaries, and downstream along the Middle and Lower Godavari subbasins (n=25). In contrast, the relative abundance of brGDGT Ia was 50% in soils from the upper basin (n=21, p≤0.001), where brGDGT Ib was the most dominant compound at a few sites instead (28-53%, n=3, sites 42, 46, 54; Fig. 2h, 4b). BrGDGT IIa' and IIIa' were the most prominent penta- and hexamethylated compounds respectively, where relative abundances of IIa' varied from 8-27% (average 17±1%, n=21) in the upper basin to 2-19%





(average 10±1%, n=25) in the lower basin (p≤0.001), with lowest values in the East Tributaries (2-9%, n=6) (Fig. 2i, 4b). The contributions of brGDGTs IIb, IIc, IIc', IIIb, IIIb', IIIc, IIIc', IIIa rarely exceeded 2%, and also IIIa' contributed mostly ≤2%. The fine fractions had similar brGDGT distributions as bulk soils (p>0.05), albeit with slightly higher relative abundances of IIa' (18±2%) at the expense of Ia (Fig. 2h, 4a,b). The relative contribution of 6-methylated isomers to brGDGTs in bulk soils

varied spatially (p≤0.001), and ranged from 26±1% (n=19) in the Upper Godavari to 21±2% (n=12) in the North Tributaries, 6±2% (n=6) in the East Tributaries, 20±2% (n=4) in the Middle, and 18±2% (n=5) in Lower Godavari. Crenarchaeol concentrations were 0.1 to 3.3 µg g$^{-1}$ OC (1.2±0.1), resulting in BIT index values of 0.52 to 0.91 (0.75±0.01), where the lowest values (<0.65) occurred in the Upper Godavari subbasin and in disturbed soils (i.e., agricultural soils or soils influenced by mining activities) from the North Tributaries and Lower Godavari (Fig. 2j).

**4.2.2 GDGTs in surface SPM**

BrGDGTs were detected in all SPM samples, but brGDGT IIIc was always below the detection limit. BrGDGTs IIc, IIIa, IIIb, IIIb' and IIIc' had relative abundances ≤ 0.01, and contributions of brGDGTs IIb and IIc' rarely exceeded 2% in both the dry and wet season (Fig. 4a).

In surface SPM collected in the dry season (n=18), brGDGT concentrations were between 1.2 and 19.6 µg g$^{-1}$ OC (average

8.1±1.2), without a distinct trend across the basin (Fig. 2d). Tetramethylated brGDGTs had the highest relative abundance (39-73%, average 54±3%), followed by penta- (25-50%, 40±2%) and hexamethylated brGDGTs (0-12%, 6±1%). BrGDGT Ia was the most abundant compound (26-53%, 37±2%), followed by IIa' (12-36%, 23±2%) and Ib (8-21%, 15±8%) (Fig. 4a,c). BrGDGT Ia generally dominated in the Lower Godavari, whereas brGDGT IIa' was particularly abundant in the Upper Godavari and North Tributaries (Fig. 2h,i, 4c). The contribution of 6-methyl brGDGTs ranged from 20 to 60% (41±3%) with

slightly higher contributions in the upper basin (47±6%, n=4) than in the lower basin (39±3%, n=14). Crenarchaeol concentrations varied from 0.2 to 10.0 µg g$^{-1}$ OC (1.4±0.5), resulting in BIT values between 0.47 and 0.96 (0.79±0.04). The BIT was significantly higher in the upper basin (0.91±0.01, n=4) than in the lower basin (0.76±0.05, n=14, p≤0.01) (Fig. 2j), and decreased downstream to ≤0.75 in the Middle and Lower Godavari.

BrGDGT concentrations in surface SPM collected in the wet season (n=40) varied between 0.8 and 28.5 µg g$^{-1}$ OC (9.7±1.0)

(Fig. 2d). The OC-normalised brGDGT concentrations were similar to that in SPM in the dry season, but almost ~2 times higher than in Godavari soils (Fig. 2d). During the wet season, OC-normalised brGDGT concentrations showed no clear spatial variability, and were on average 7.7±1.4 µg g$^{-1}$ OC in the Upper Godavari (1.9-15.1, n=12), then slightly increasing downstream to the Middle (8.5±1.2, 5.9-11.8, n=5) and Lower Godavari (11.2±2.7, 2.0-28.5, n=10), while concentrations were 13.0±1.8 µg g$^{-1}$ OC in the North (6.3-20.1, n=9) and 6.6±3.1 µg g$^{-1}$ OC in the East Tributaries (0.8-14.9, n=4). Wet season

SPM was dominated by tetramethylated brGDGTs (50-85%, average 68±2%), followed by penta- (15-45%, 29±1%) and hexamethylated (0-6%, 3±0%) brGDGTs. In contrast to the total concentration of brGDGTs, their relative distribution did vary spatially. For example, the average contribution of brGDGT Ia was significantly lower in the upper than in the lower basin (40±1% vs 61±2%, p≤0.001) (Fig. 2h), whereas brGDGT IIa' showed an opposite trend from 26±1% in the upper to 14±1% in





the lower basin (p≤0.001) (Fig. 2i). Similarly, the contribution of 6-methyl isomers was significantly higher in the upper

(40±1%, n=14) than in the lower basin (21±1%, n=26; p≤0.001), following the trend in Godavari soils. The average

contribution of 6-methyl brGDGTs in wet season SPM was 28±2%, which is lower than in the dry season. Crenarchaeol

concentrations varied from 0.1 to 5.4 µg g⁻¹ OC (1.6±0.2), resulting in BIT values between 0.62 and 0.96 (0.84±0.01). The

BIT was significantly higher in the upper basin (0.87±0.02) than in the lower basin (0.83±0.01, p≤0.05) (Fig. 2j), similar to

dry season SPM.

**4.2.3 GDGTs in SPM depth profiles**

In SPM depth profiles from the Godavari delta (site 10, Fig. 1c), brGDGT concentrations varied from 1.2 to 4.0 µg g⁻¹ OC in

the dry season (n=3, 3 depths for 1 profile at mid-river position), and from 8.5 to 31.8 µg g⁻¹ OC in the wet season (n=9, 3

depths for 3 profiles across the river), and showed no trend with depth (Fig. 3d, solid lines). BrGDGT Ia was the most abundant

compound, contributing 39-44% in the dry, and 66-69% to the total amount of brGDGTs in the wet season.


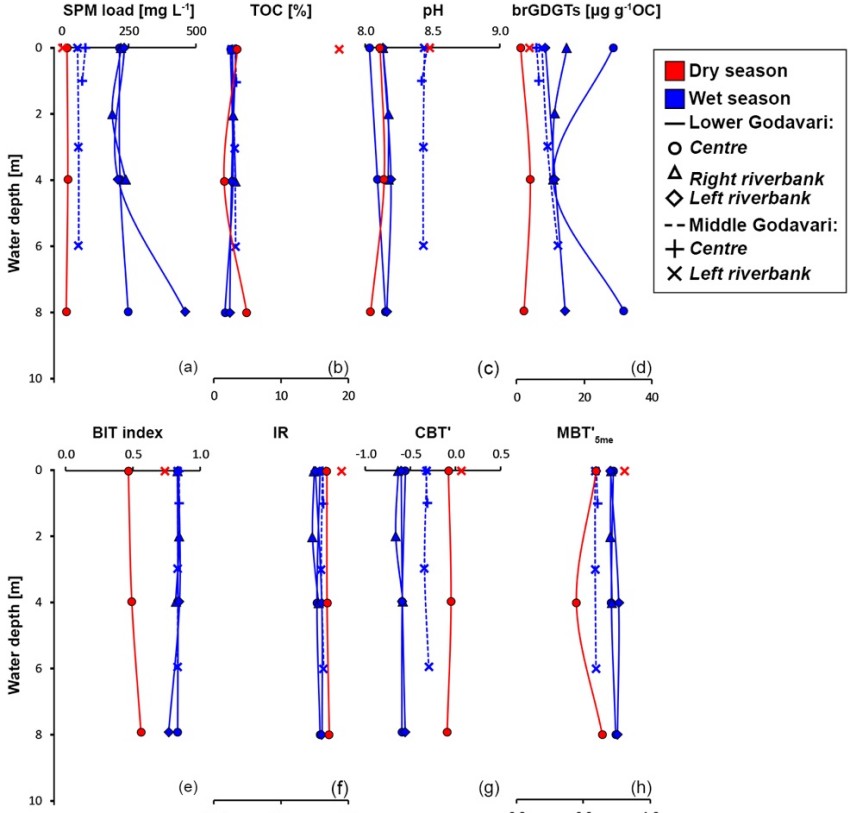

**Figure 3 – River depth profiles of bulk properties, (a) Suspended Particulate Matter (SPM) load, (b) Total Organic Carbon (TOC), (c) measured pH in the water column, (d) brGDGT concentration, and of brGDGT-based indices, (e) BIT index, (f) Isomer Ratio (IR), (g) CBT' and (h) MBT'₅ₘₑ index values. The symbols represent the position in the channel: mid-channel (centre), and near the**
**left and right riverbank. In the Lower Godavari, the right is the non-eroding and the left is the eroding riverbank.**





The relative abundances of individual brGDGTs showed minor variation with depth in both seasons. The contribution of 6-methyl brGDGTs varied between 30 and 37% in the dry, and 14 and 17% in the wet season. In the Middle Godavari (site 28; Fig. 1b), SPM depth profiles were only collected in the wet season. There, brGDGT concentrations ranged from 5.9 to 12.2

µg g$^{-1}$ OC (n=5) (Fig. 3d, dashed lines), with brGDGT Ia as most abundant compound (53-55%). The contribution of 6-methyl brGDGTs was 25-27%.

### 4.2.4 GDGTs in riverbed sediments

BrGDGTs were detected in all bulk riverbed samples collected in the dry (n=37) and wet season (n=37), as well as in the fine fraction of the sediments sampled in the wet season (n=25). Like in the Godavari soils, the contribution of brGDGTs IIb, IIc,

IIc', IIIa, IIIb, IIIb', IIIc and IIIc' rarely exceeded 2% (Fig. 4a).

In the dry season, brGDGT concentrations ranged from 2.4 to 85.0 µg g$^{-1}$ OC (16.6±2.4), with higher concentrations in the upper (26.8±5.7, n=13) than in the lower basin (11.1±1.1, n=24, p≤0.02) (Fig. 2d). Tetramethylated brGDGTs often dominated (relative abundance 40-78%, average 59±2%), followed by penta- (20-50%, 36±2%) and hexamethylated (2-11%, 5±0%) brGDGTs. BrGDGT Ia was generally the most abundant compound (18-62%, 39±2%) and its relative abundance was

significantly lower in the upper (30±1%) than in the lower basin (44±2%, p≤0.001) (Fig. 2h, 4a,d). BrGDGT IIa' showed an opposite spatial trend and was significantly more abundant in the upper basin (23±2%) than in the lower basin (15±1%, p≤0.001) (Fig. 2i). The contribution of 6-methyl isomers was higher in the upper (43±2%) than the lower basin (30±2%, p≤0.001). This contribution varied spatially (p≤0.001) from 43±2% (n=11) in the Upper Godavari, 47±2% (n=6) in the North and 28±2% (n=3) in the East Tributaries to 40±4% (n=4) in the Middle, and 22±2% (n=13) in the Lower Godavari.

Crenarchaeol concentrations ranged from 0.0 to 14.5 µg g$^{-1}$ OC (2.7±0.5), translating into BIT values of 0.51-1.00 (0.80±0.02). The upper basin had a higher average BIT value (0.86±0.02) than the lower basin (0.76±0.02, p≤0.01) (Fig. 2j).

In the wet season, OC-normalised brGDGT concentrations in bulk sediments were on average ~2 times higher than in Godavari soils, and similar to concentrations in wet season SPM. BrGDGT concentrations varied between 1.2 and 36.9 µg g-1 OC (12.5±1.6, n=37) and were higher in the upper (18.0±3.8, n=8) than in the lower basin (11.0±1.6, n=29) (Fig. 2d). Fine fraction

sediments had up to 10 times higher brGDGT concentrations than the bulk sediments from the same site (22.1±1.4 vs 13.1±1.7 µg g$^{-1}$ OC, p≤0.001, n=25). The relative abundances were similar for fine and bulk sediments (Fig. 4a), and are dominated by tetramethylated brGDGTs (47-86%, 68±2%), followed by penta- (13-46%, 29±1%) and hexamethylated (0-9%, 3±0%) compounds, similar to in Godavari soils (Fig. 4a). BrGDGT Ia was the most abundant compound (21-77%, 46±2%), except for site 42, where Ib dominated (Fig. 4a,d). The contribution of brGDGT Ia was substantially higher than in the dry season

(p≤0.02), and generally increased downriver, resulting in a significantly lower relative abundance in the upper (33±3%, n=8) than in the lower basin (50±2%, n=29, p≤0.001) (Fig. 2h). BrGDGT IIa' showed the opposite spatial trend and had a higher relative abundance in the upper basin (21±2%) than in the lower basin (13±1%, p≤0.01) (Fig. 2i). The contribution of 6-methyl isomers showed a similar trend as in Godavari soils and wet season SPM, varying spatially (p≤0.001) from 38±3% in the





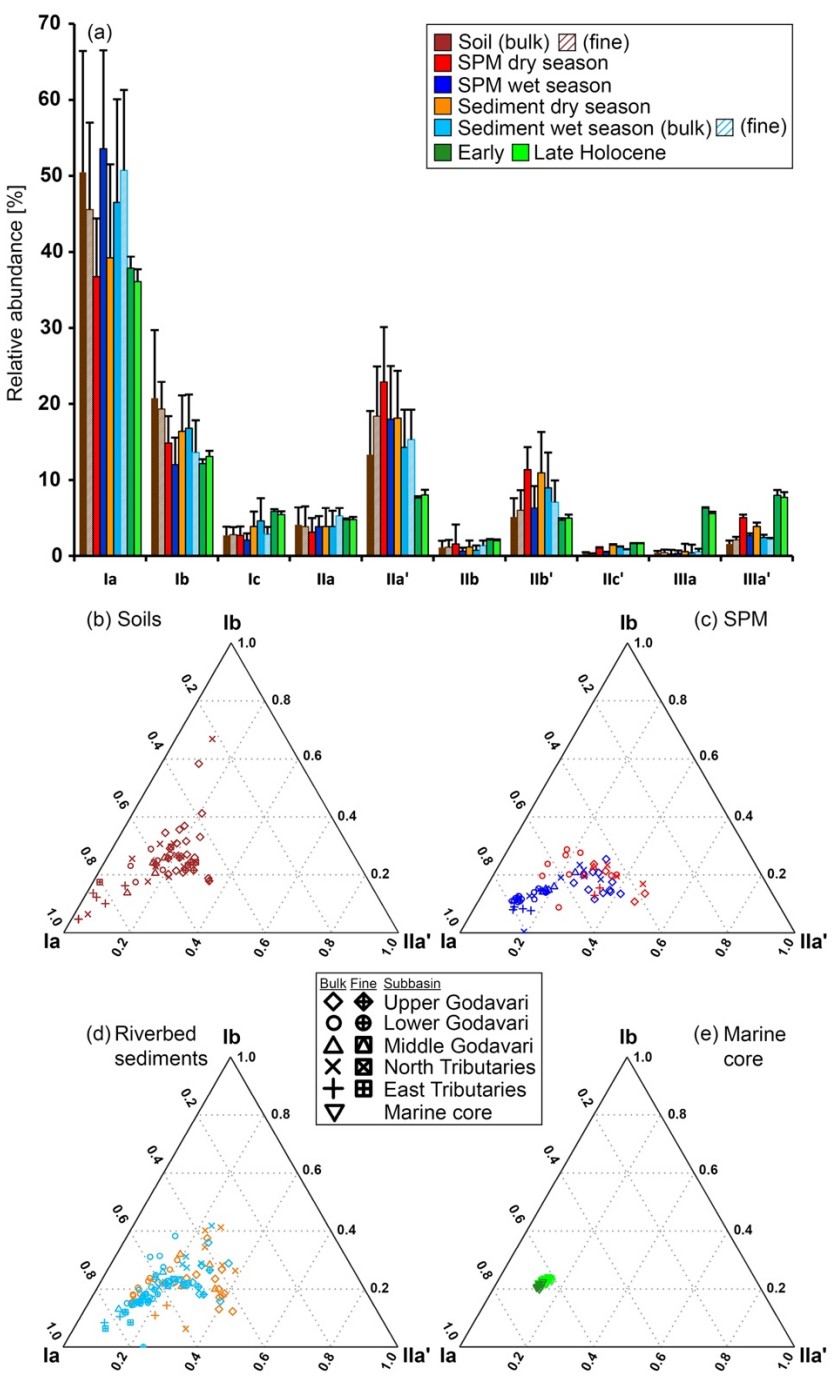

**Figure 4 – BrGDGT distributions in the Godavari basin and in marine sediments from the Bay of Bengal. (a) Average relative abundances of brGDGTs in soils, SPM and riverbed sediments in the dry and wet season, and in Holocene marine sediments. The error bars represent the standard deviation. The fine fraction soils and sediments (≤63 μm) are striped. Proportions of major brGDGTs Ia, IIa' and Ib in (b) soils, (c) SPM, (d) riverbed sediments and in a (e) Holocene marine core. The symbols represent the different subbasins, with open symbols for the bulk soils and sediments and enclosed symbols for the fine fractions.**





Upper Godavari to 35±2% in the North Tributaries, 13±2% in the East Tributaries, and to 26±4% in the Middle, and 22±1%
in the Lower Godavari. Notably, contributions of 6-methyl brGDGTs were on average lower than in the dry season (p≤0.01).
At the site in the delta where depth profiles were collected (site 10), brGDGT concentrations in sediments were lowest near
the non-eroding riverbank. Also, the contributions of Ib and Ic were relatively higher at the expense of Ia, compared to the
middle of the river and close to the eroding riverbank. The cross-section sampled in the Middle Godavari (site 28) showed

marginal variation in both brGDGT concentration and distribution. Crenarchaeol concentrations in wet season bulk sediments
ranged from 0.1 to 6.6 µg g$^{-1}$ OC (2.0±0.2), resulting in BIT index values of 0.44-0.93 (0.78±0.02) (Fig. 2j). BIT index values
showed no distinct spatial trend but were remarkably high in the East Tributaries (0.88-0.92, n=3). For fine fraction sediments,
crenarchaeol concentrations varied between 0.1 and 13.1 µg g$^{-1}$ OC (4.4±0.5), translating into BIT values of 0.64-0.93
(0.79±0.02), with the same trend as for the bulk sediments.

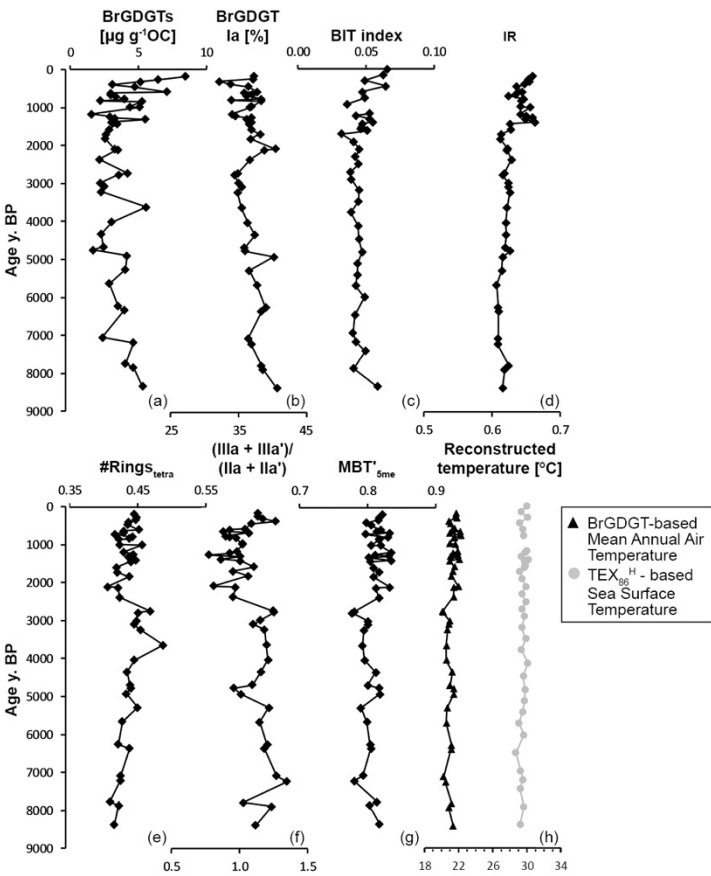


**Figure 5 – Branched and isoprenoid GDGTs and their indices in Holocene marine sediments of core NGHP-01-16A (~1250 m water depth), retrieved ~40 km from the Godavari River mouth in the Bay of Bengal. The age model from Usman et al. (2018) is used. (a) Concentration of brGDGTs, (b) relative abundance of Ia, (c) BIT index, (d) IR, (e) #rings$_{tetra}$, (f) (IIIa+IIIa')/(IIa+IIa') (following Xiao et al., 2016, 2020), (g) MBT'$_{5me}$ index values and (h) reconstructed temperatures, (black triangles): continental MAAT based**
**on the MBT'$_{5me}$ index and using the BayMBT$_0$ model of Dearing Crampton-Flood et al., (2020), and (grey circles): Sea Surface Temperature, based on the TEX$_{86}^H$ index, following Kim et al. (2010). Note that the BIT index and SST records are at a slightly lower resolution (n=35) than the brGDGT records (n=46) due to the use of a different set of lipid extracts.**





### 4.2.5 GDGTs in Holocene Bay of Bengal sediments

All brGDGTs were detected in marine sediment core NGHP-01-16A spanning the Early to Late Holocene period (n=46) (Fig.
4a) (Kirkels et al., 2021b). However, the contributions of IIc, IIIb, IIIb', IIIc and IIIc' were always <1% and also IIb and IIc'
contributed mostly <2% to the total brGDGT pool. The distribution of individual brGDGTs was mostly similar with depth,
but showed a slightly higher variability in the upper part of the core corresponding with the past ~2000 years (Fig. 5). In
general, tetramethylated brGDGTs were the most abundant compounds (relative abundance 49-59%), followed by penta- (22-
31%) and hexamethylated brGDGTs (11-19%). BrGDGT Ia (32-40%, average 37±0%) was the most abundant compound (Fig.
2h, 4a,e, 5b). BrGDGT concentrations varied from 1.6 to 8.4 µg g$^{-1}$ OC (3.7±0.2), without a clear trend with depth (Fig. 2d,
5b). The contribution of 6-methyl isomers ranged from 19 to 26% but was relatively invariant with depth. Crenarchaeol
concentrations were not quantified, as the amount of sediment used for extraction was not determined for the samples analysed
for isoGDGTs (n=38), but BIT index values ranged from 0.03 to 0.07 (Fig. 2j, 5c).

## 5 Discussion

### 5.1 Spatial variations in GDGTs in Godavari soils

To determine the provenance of brGDGTs in the Godavari River and of those exported to the Bay of Bengal, we first investigate
spatial variations in brGDGT distributions in surface soils across the basin. BrGDGT Ia, which is typically associated with
high temperatures in tropical to semi-arid regions (Weijers et al., 2007a; De Jonge et al., 2014a), is by far the most abundant
compound in Godavari soils (Fig. 4a). The relative abundance of brGDGT Ia is higher in the upper than in the lower basin
(Fig. 2h). Given the minor temperature variation across the Godavari basin (<3.5 °C), this suggests that its abundance is driven
by another parameter than temperature. The relative abundances of all other brGDGTs vary considerably across the basin, but
do not show a clear spatial trend.

A principal component analysis (PCA) of brGDGTs in Godavari soils reflects the high variability in brGDGT distributions
across the basin, as soils from the different subbasins are not restricted to single quadrants (Fig. 6a). Nevertheless, PC1 explains
50% of the total variance, and broadly separates soils from the East Tributaries that load positive on PC1 with high relative
abundances of brGDGT Ia, from soils in the Upper Godavari and North Tributaries that have a negative loading on this PC,
and contain higher abundances of penta- and hexamethylated brGDGTs, whereas soils from the Middle and Lower Godavari
plot in the middle. Interestingly, most soils that plot negative on PC1 are formed on the Deccan basalts that underlie the upper
basin, while those with positive loadings have developed on the felsic rocks in the lower basin. However, this separation is not
consistent enough to link the different bedrock types to certain brGDGTs, indicating that brGDGT signatures alone cannot be
used to trace basin-specific contributions in the Godavari basin.





Although the BIT index is generally used to determine the relative contribution of terrestrial OC in a marine system, this

index is sensitive to moisture availability in soils. Archaea that produce isoprenoid GDGTs, including crenarchaeol, appear

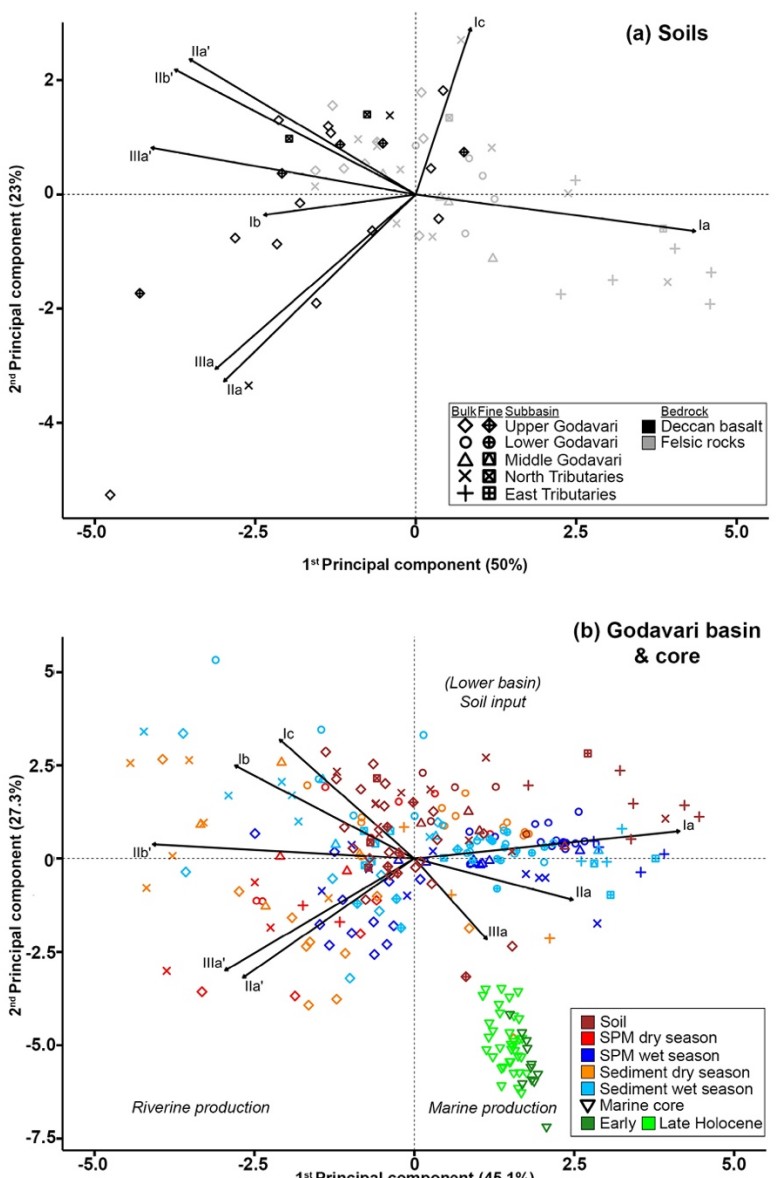

to be

**Figure 6 – Principal Component Analysis (PCA) of the eight major brGDGTs (i.e., Ia, Ib, Ic, IIa, IIa', IIb', IIIa and IIIa'). (a) PCA for soils formed on Deccan basalts (black) and felsic bedrocks (grey) (PC1 versus PC2). (b) PCA for SPM and bulk riverbed sediments collected in the wet and dry season (PC1 versus PC2), to which PCA scores of the bulk soils and Holocene marine sediments are added passively. The latter implies that the PCA calculation and biplot configuration is based on brGDGT distributions in river SPM and sediments (i.e., 'active' samples), to which the scores of soils and marine sediments (i.e., 'passive' samples) are added as an overlay. The vectors indicate the PCA scores of the individual brGDGTs. The symbols refer to the different subbasins and the different fractions (bulk and fine fractions ≤63 µm) (see legend a). The colours refer to the different sample types.**





better resistant to arid conditions than the bacteria that produce brGDGTs, as previously shown for dry, alkaline soils in China
(Xie et al., 2012; Yang et al., 2014) and North-America (Dirghangi et al., 2013). Hence, the low BIT values in Godavari soils
(0.52-0.91), especially in the upper basin, can be explained by the semi-arid to arid climate and slightly alkaline nature of the
soils. However, there is no spatial trend in the BIT index across the basin (Fig. 2j), indicating that also the BIT index cannot
be used to determine the provenance of brGDGTs carried by the Godavari River.

## 5.2 Sources of GDGTs in the Godavari River

To identify the areas of soil OC input into the Godavari River and the influence of (hydro)climate conditions under which soil
OC is mobilised, the soil brGDGT distributions are compared with those in SPM and riverbed sediment collected in the dry
and wet season. In contrast to the Godavari soils, a PCA of the relative abundances of eight major brGDGTs in SPM and bulk
riverbed sediments collected in both seasons does reveal trends in brGDGT composition that can be linked to spatial and
seasonal changes (Fig. 6b). PC1, explaining 45.1% of the total variance, clearly separates acyclic 5-methyl brGDGTs Ia, IIa
and IIIa from 6-methyl brGDGTs IIa', IIb' and IIIa', as well as the upper and the lower basin. This suggests that the microbial
community shifts from more 6-methyl brGDGT producers in the upper basin, to a more 5-methyl brGDGT producing
community in the lower basin. PC2, explaining 27.3% of the total variance, seems to separate both the degree of methylation
of brGDGTs, as well as the degree of cyclisation, where brGDGTs without additional methylations but with one or two
cyclopentane moieties plot positive on PC2 (Fig. 6b). This PC further tears apart SPM and riverbed sediments collected in the
upper basin that have high relative abundances of brGDGT IIa' and IIIa', from those collected in the lower basin in the wet
season which are distinct by high relative abundances of tetramethylated brGDGTs. Together, PC1 and PC2 show that SPM
and riverbed sediments from the dry season as well as those collected in the upper basin are characterised by high relative
abundances of 6-methyl brGDGTs, whereas SPM and sediments collected in the wet season in the lower basin are associated
with more 5-methyl brGDGTs and brGDGT Ia.

Passively adding the soil brGDGTs distributions to the PCA biplot based SPM and sediments shows that the soils mostly plot
in the upper two quadrants, and generally overlap with SPM and sediments collected in the lower basin in the wet season (Fig.
6b). Specifically, soils developed on felsic bedrocks in the lower basin plot in the right upper quadrant, together with SPM and
sediments collected in the same part of the basin in the wet season. This similarity suggests that brGDGTs in this part of the
Godavari River are most likely soil-derived. In contrast, the soils from the Deccan plateau plot opposite to both wet and dry
season SPM and sediments collected from the same region on PC2. This difference is mainly driven by higher relative
abundances of brGDGTs IIa' and IIIa' in SPM and sediments versus higher abundances of Ib and Ic in Deccan soils. This
opposite loading on PC2 suggests that there is limited transfer of Deccan soils into the river, or that soil-derived brGDGTs are
overprinted by brGDGTs from other sources.

In general, SPM and sediments collected throughout the Godavari basin in the dry season, as well as those from the upper
basin year-round, show relatively limited overlap with the soils (Fig. 6b). These samples are characterised by higher relative





abundances of 6-methyl brGDGTs, and brGDGT IIa' in particular, but lower abundances of Ia compared to soils (Fig. 2h,i, 4a-d). Based on the offset between brGDGT distributions in soils and SPM in the Lower Amazon River, collected during low flow conditions when soil input is limited, Zell et al. (2013a,b) suggested that brGDGTs could also be produced in situ in the river. Similar offsets were also found in the Yangtze (Yang et al., 2013; Li et al., 2015), Rhône (Kim et al., 2015) and Tagus
(Zell et al., 2014a) rivers. Later, in situ aquatic brGDGT production was linked to the occurrence of 6-methyl brGDGTs, based on their unexpectedly high relative abundance during low flow conditions in the Yenisei River (De Jonge et al., 2014b). Also sediments and SPM from quiescent, less turbid waters in front of the Iron Gates in the Danube River (Freymond et al., 2017) and the upper Amazon River in the dry season (Kirkels et al., 2020a) were characterised by high contributions of 6-methyl brGDGTs compared to in soils, quantified in high IR values (Eq. 3). In the Godavari River, IR values in dry season SPM
(0.88±0.02, p≤0.001) and riverbed sediments (0.85±0.01, p≤0.001), as well as SPM (0.90±0.01, p≤0.001) and sediments (0.89±0.02, p≤0.05) from the upper basin in both seasons are significantly higher than those in soils (0.76±0.02), suggesting that the brGDGTs in these subbasins and seasons are mainly produced in situ in the river (Fig 2k, 7).

Recently, mesocosm experiments confirmed that brGDGTs, and 6-methyl brGDGTs in particular, are produced in situ in the water column, especially at high(er) temperatures, high nutrient levels and low oxygen conditions (Martínez-Sosa and Tierney,
2019; Martínez-Sosa et al., 2020). Similar conditions prevail in the Godavari basin in the dry season, when water temperatures are high (~29°C; Kirkels et al., 2020b), and nutrient inputs from agriculture and wastewater effluents facilitate phytoplankton (blooms). The subsequent degradation of algal biomass consumes oxygen (Pradhan et al., 2014), creating the low oxygen conditions favourable for brGDGT production. Low suspended sediment loads and relatively high %OC in the Godavari River in the dry season (Fig. 2a, b) suggest that soil input is low, and waters are non-turbid, further facilitating primary (autotrophic)
and secondary (heterotrophic: e.g., bacteria) aquatic production. Indeed, Kirkels et al. (2020a) reported that a decrease in turbidity in the dry season in an upper Amazon tributary promoted increased production of heterotrophic brGDGT-producing bacteria. A similar scenario has been described to explain brGDGT production behind dams in the Yangtze (Yang et al., 2013) and Danube (Freymond et al., 2017) rivers where the reduced flow velocity and turbidity facilitated aquatic brGDGT production. Notably, OC-normalised brGDGT concentrations in Godavari riverbed sediments are ~2 times higher than in SPM
in the dry season, especially in the upper basin (Fig. 2d). This suggests that brGDGTs may also be produced in the sediment, although the overlap of SPM and sediments from the upper basin in the PCA biplot (Fig. 6b) could also be interpreted as preservation of aquatic produced brGDGTs in the sediment layer.

In contrast to the dry season, brGDGT distributions in wet season SPM and riverbed sediments from the lower basin resemble that of Godavari soils (Fig. 2h, 4a-c). This suggests that that soil OC is mobilised and transferred into the Godavari River in
the lower basin. Indeed, the IR of wet season SPM (0.80±0.01) and sediments (0.81±0.02) from the lower basin is lower than in the dry season (SPM: 0.88±0.02; sediment: 0.84±0.02) and instead resembles that of soils (0.76±0.02) (Fig. 2k, 7). In contrast, the IR remains significantly higher for wet season SPM and sediments in the upper basin (p≤0.05), suggesting that brGDGTs in the upper basin have a predominantly aquatic source year-round. The change in brGDGT sources from the upper to the lower basin in the wet season reflects the precipitation pattern that was established based on water isotopes of Godavari



riverwater in the year of sampling (Kirkels et al., 2020b). The lower basin, and especially the North and East Tributary regions, received most precipitation thus promoting soil mobilisation, whereas the upper basin experienced a severe precipitation deficit that hampered soil mobilisation and downstream transport (Fig. 1d). In addition, abundant dams in the upper basin reduce the river flow and create stagnant waters that facilitate aquatic productivity year-round (Pradhan et al., 2014; Kirkels et al., 2020b). The in situ production of brGDGTs in the upper basin, especially that of 6-methyl brGDGTs IIa' and IIIa', results in

significantly higher BIT index values for both wet (0.87±0.02, p≤0.05) and dry season (0.91±0.04, p≤0.01) SPM compared to for the lower basin (wet: 0.83±0.01; dry season: 0.76±0.05) (Fig. 2j). This contrasts with whitewater rivers carrying high suspended sediment loads in the Lower Amazon basin, where BIT index values were found to decrease in the dry season as a result of crenarchaeol production in the river (Kim et al., 2012; Zell et al., 2013a). A possible explanation for the opposite trend in BIT index is that the production of crenarchaeol by ammonia oxidising archaea in the Lower Amazon depends on

phytoplankton blooms to release N, whereas N levels are continuously high in the upper basin of the Godavari River (Gupta et al., 1997; Pradhan et al., 2014). Hence, it seems that the presence of stagnant waters and high nutrient levels may be important factors in determining whether conditions are favourable for the production of (6-methyl) brGDGTs or crenarchaeol in rivers, and thus affect the BIT index.

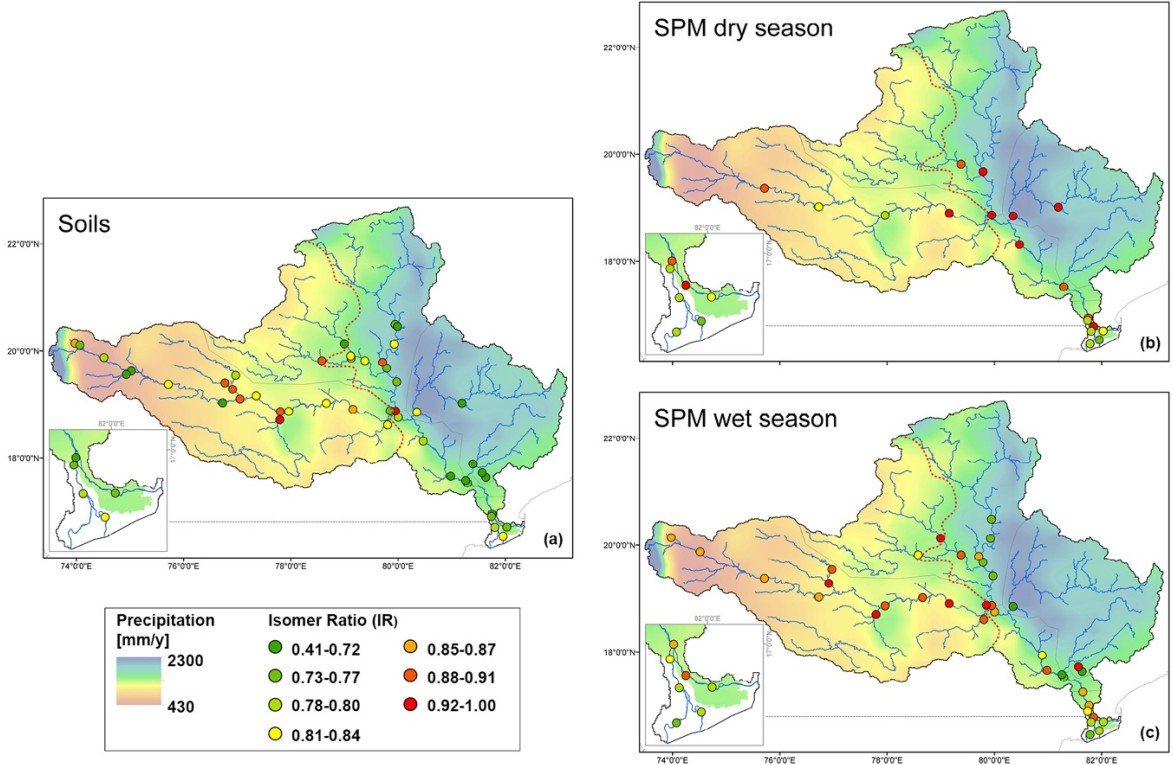

**Figure 7 – Maps showing the spatial distribution of IR values for (a) soils and SPM collected in the (b) dry and (c) wet season. The main panel shows the whole basin, with a zoom for the delta region. The coloured points refer to the IR values and the long-term average rainfall distribution is shown on the background.**



Interestingly, prior to reaching the river mouth, the seasonal variation in brGDGT sources and distributions seems to be partly smoothened by the Rajahmundry Reservoir Lake that controls the discharge to the Godavari delta. The SPM and sediments

collected downstream of the reservoir all plot positive on PC2 (Fig. 6b), representing a mixture of soil-derived brGDGTs (characterised by Ia) transported from the North and East Tributaries in the wet season and aquatically produced brGDGTs (6-methyl brGDGTs) from the dry season.

### 5.3 Modes of soil OC transport through the modern Godavari basin

### 5.3.1 Sediment provenance

In order to assess whether brGDGTs are transported free or associated to mineral surfaces, we first determine the elemental composition of mineral particles in soils and riverbed sediments across the Godavari basin. The two distinct lithological units of the Godavari basin, i.e., Deccan basalts mainly in the upper basin and the felsic metamorphic and sedimentary rocks in the lower basin (Fig. 1e), are reflected in the elemental composition of the soils. Concentrations of both Ti and Fe are significantly higher in soils from the Deccan region than in the felsic region. This difference can be explained by the presence of Ti- and

Fe-oxydydrates in the Deccan basalts. In contrast, K concentrations are significantly higher in the felsic region, linked to the abundance of K-feldspars in these felsic bedrocks. The transition in bedrock composition is reflected in the trend in Ti/K and Fe/K ratios of the soils (Fig. 8a, Appendix 2), and match known end-member values for the Deccan basalts (Ti/K: 2.79, Fe/K: 18.13; Das and Krishnaswami, 2007) and felsic bedrocks (Ti/K: 0.21; Fe/K: 1.10; Moyen et al., 2003) in the basin. This implies that these ratios can be used to trace sediment provenance in the Godavari basin. Ti/K and Fe/K ratios have previously been

used to reconstruct hydroclimate-related changes in chemical weathering intensity in the Mekong (Jiwarungrueangkul et al., 2019), Nile (Bastian et al., 2017) and Zambezi (Just et al., 2014) river basins, as well as to trace Andean inputs in modern riverbed sediments of the Lower Amazon River (Häggi et al., 2016), its proximal fan (Govin et al., 2012, 2014) and in shelf sediments along the Chilean coast (Stuut et al., 2007). In the Godavari basin, Ti/K and Fe/K ratios of bulk sediments clearly decrease from the Deccan basalts region to the lower basin (felsic bedrocks) (Fig. 8b,c, Appendix 2). Notably, the bulk

sediments in the dry and wet season show an abrupt change in elemental ratio values to the felsic end-member ~700 km from the river mouth. The fact that ratios are broadly similar for soils and riverbed sediments from the same location suggests that the sediments have a predominantly local/regional provenance. This implies that Deccan-derived material is not transported to the lower basin (Fig. 8b,c), probably as result of sediment trapping by the abundant dams in the upper basin and limited rainfall in the year of sample collection. The local, felsic origin of riverbed sediments in the lower basin and delta of the

modern Godavari River is also confirmed by their neodymium isotopic composition (εNd; Ahmad et al., 2009). In addition, Ti/K and Fe/K ratios in marine cores covering the past ~300 yr taken in front of the Godavari River mouth (Kalesha et al., 1980) closely match the end-member values for felsic bedrocks as well as those of the modern riverbed sediments in the lower basin (Fig. 8b,c, Appendix 2). Taken together, this indicates that the felsic bedrock region has been the dominant source of fluvial sediment delivered to the adjacent continental margin over the last centuries.




**Figure 8** – Ti/K ratios in (a) soils and riverbed sediments collected in the (b) dry and (c) wet season versus distance to/from the river mouth. The square and star (magenta) represent the end-member values for the Deccan basalts (Das and Krishnaswami, 2007) and felsic bedrocks (Archean Proterozoic Gneiss Complexes, APGC) (Moyen et al., 2003), respectively. The outlined star (greenyellow)

represents the average of marine sediments retrieved from * shallow cores in front of the Godavari mouth (0-48 cm below sea floor, ~300 yr.; Kalesha et al., 1980) (error bars representing the standard deviation (±0.02) are not visible as they are smaller than the symbol). %OC and brGDGT concentrations versus Al/Si ratios in soils (d, g) and riverbed sediments collected in the (e, h) dry and (f, i) wet season, respectively. Low Al/Si ratios indicate a bulk mineralogical composition with high proportions of quartz (coarse grained) and high ratios indicate high proportions of micas and clays (fine grained) (Galy et al., 2008, 2010). Linear correlations

(solid line) and 95% confidence intervals (grey shading) are shown. The symbols refer to the different subbasins, and the different fractions (bulk and fine fractions ≤63 μm). The colours refer to the different sample types.

## 5.3.2 Mineral associations during river transport

To investigate possible association of OC and brGDGTs with mineral particles, as well as their downstream evolution, %OC and brGDGT signals are normalised to the Al/Si ratio. This normalisation helps to account for the change in bedrock geology





in the Godavari basin, which may result in differences in grain size upon weathering (Fig. 8d-i, Appendix 3). Al is typically

enriched in clays, which are also characterised by a high surface area. On the other hand, felsic bedrock weathers into Si-

enriched, coarse-grained, low mineral surface area material. The Al/Si ratio can thus serve as a proxy for the abundance of

fine-grained, high surface area aluminosilicates that can host OC (Galy et al., 2008, 2010). For example, strong physical

associations between OC and minerals were inferred from positive linear correlations between %OC and Al/Si in the Amazon

and Ganges-Brahmaputra Rivers (Galy et al., 2007, 2008, 2010; Bouchez et al., 2014; Häggi et al., 2016). This positive

correlation revealed that mineral particles and OC respond in the same way across a range of bedrock types and seasonally

contrasting hydrological conditions. At a molecular level, a similarly positive correlation between lignin concentrations and

Al/Si in the Amazon basin showed that lignin was preferentially associated with fine-grained sediments (Sun et al., 2017). In

contrast, n-alkanes that are also derived from vascular plants and were extracted from the same material did not show this

correlation, which was explained by different source areas of n-alkanes and mineral particles (Häggi et al., 2016). In addition,

the different behaviour of lignin and n-alkanes in the same river system suggests that molecular level OC sorting may take

place during land-sea transfer, possibly depending on the properties of the molecule.

In the Godavari basin, the %OC of dry season riverbed sediments only shows a weak trend with Al/Si ($R^2$ = 0.48, Fig. 8e),

indicating that OC was not closely associated with mineral particles. In the wet season, the %OC of riverbed sediments does

show a positive linear correlation with Al/Si ($R^2$ = 0.71, Fig. 8f), revealing that bulk OC and mineral particles are transported

together, possibly as a result of close OC-mineral association. A similar relation between brGDGT concentrations and Al/Si

ratios would reveal if brGDGTs are also associated with mineral particles during river transport, which would make them

robust tracers for soil OC. Like bulk OC, brGDGT concentrations in riverbed sediments show a weak relation with Al/Si in

the dry season ($R^2$=0.54), excluding one outlier (site 47, in the arid, upper basin) that had exceptionally high brGDGT

concentrations (Fig. 8h). BrGDGT concentrations in riverbed sediments collected in the wet season reveal a somewhat stronger

correlation with Al/Si ($R^2$=0.60) (Fig. 8i), although this correlation is weaker than that for bulk OC and Al/Si (Fig. 8f, i). This

difference suggests that brGDGTs may be less (strongly) associated with mineral particles than bulk OC. Indeed, earlier studies

have suggested that brGDGT-mineral associations are continuously renewed and/or replaced by aquatic production or by local

inputs during river transit (Li et al., 2015; Freymond et al., 2017, 2018b). Partitioning of brGDGTs into colloids, where they

are dispersed in the riverwater but not directly associated with mineral surfaces, or preferential degradation of brGDGTs over

bulk OC could offer an alternative explanation for the slightly different trend of brGDGTs and bulk OC in the Godavari basin.

Given the hydrophobic nature of brGDGTs, transition into the dissolved phase is less likely.

### 5.3.3 Linking sediment and brGDGT transport

Following brGDGTs and mineral elemental ratios, both the mineral particles and the brGDGTs delivered to the modern-day

river mouth appear to be sourced from the lower part of the Godavari basin. This is in contrast with other large river systems,

where the provenance of these biomarkers and mineral particles seems more often decoupled. For instance, in the Amazon

River, exported mineral fractions had an Andean signature, whereas brGDGTs in the Lower Amazon and in the river fan





sediments were sourced from lowland soil inputs and in situ production (Zell et al., 2013a, 2014b; van Soelen et al., 2017; Kirkels et al., 2020a). Similarly, in the Yangtze (Li et al., 2015) and Danube rivers (Freymond et al., 2018b), the upstream

signature of the mineral fractions was preserved, while soil-derived brGDGTs seemed to be degraded in the river and then replaced by in situ produced brGDGTs and/or local soil inputs with a different brGDGT distribution.

The observed link between brGDGTs and sediment transport in the modern-day Godavari basin may be attributed to the rainfall distribution in the wet season that mobilises soils from the lower basin and rapidly transports them downstream. The extreme turbidity and high flow velocities that characterise the Godavari River in the wet season furthermore limit autochthonous

aquatic biological activity and in-river degradation (Gupta et al., 1997; Balakrishna and Probst, 2005; Syvitski and Saito, 2007). These conditions would likely hinder brGDGT production in the river and thus prevent overprinting of the soil-derived brGDGTs. Nevertheless, as soil brGDGT distributions do not significantly vary across the lower basin, it is not possible to fully exclude brGDGT replacement by local soil inputs during downstream transport. Mobilisation of sediment and associated brGDGTs from the upper basin is limited by the below-average rainfall in this area (Kirkels et al., 2020b) and abundant dams

hindering downstream transport.

### 5.3.4 Sorting effects in the Godavari River

Hydrodynamic sorting effects within the river have been shown to result in a coarsening of SPM toward the riverbed, thereby affecting the depth distributions of OC (Goñi et al., 2005; Galy et al., 2008; Bouchez et al., 2014; Guinoiseau et al., 2016) and lipid biomarkers (Kim et al., 2012; Feng et al., 2016; Feakins et al, 2018) that are associated with certain size fractions.

Moreover, coarser and heavier mineral particles are preferentially deposited during sediment settling, favouring the transport of finer grained sediments further downstream. Comparison of the %OC in the bulk and fine fraction sediments collected in the wet season indicates that most OC is associated with the fine fraction in the Godavari sediments and may thus be susceptible to hydrodynamic sorting. Nevertheless, the %OC of SPM collected from several depth profiles in the Middle Godavari (site 28) and the delta (site 10) is relatively invariant with depth (Fig. 1b,c, 3b). This finding is in contrast to other monsoonal rivers

such as the Amazon (Bouchez et al., 2014; Kirkels et al., 2020a) and the Ganges-Brahmaputra River (Galy et al., 2008) where extensive size-related sorting is observed in the water column. In the Godavari River, only a slight drop in %OC of SPM (~1% from the upper water column to close to the riverbed) occurred at mid-channel position and near the eroding river bank in the delta in the wet season, which suggests that occasional sorting may take place at peak discharge, albeit minor.

Like bulk OC, brGDGT concentrations in riverbed sediments are higher in fine fraction than in bulk sediments, even when

normalised to %OC (Fig. 2b,d). The higher OC-normalised brGDGT concentrations in the fine fraction indicate that the grain size is important for the quantity of brGDGTs that is transported. However, brGDGT distributions and calculated proxy values are similar for fine fraction and bulk sediments, suggesting that there is no selective sorting of certain brGDGTs (Fig. 2, 4). This observation confirms previous findings that brGDGT distributions are largely uniform among different size fractions in globally distributed river sediments (Peterse and Eglinton, 2017). Depth profiles of river SPM provide further support for the

absence of sorting, as the concentration of brGDGTs, also when normalised to %OC, shows no pronounced trend with depth



(Fig. 3d). Also, brGDGT distributions, here quantified in the IR, CBT' and MBT'$_{5me}$ indices, are generally constant with depth and across the river in both the dry and wet season (Fig. 3f-h), similar to recent studies in the Danube (Freymond et al., 2018a) and the upper Amazon (Kirkels et al., 2020a). The homogeneous brGDGT distributions in sediment size fractions and depth profiles indicate that there is no hydrodynamic sorting effect on brGDGTs and no preferential transport (in the wet season) or

production (in the dry season) of individual brGDGTs at a certain depth within the Godavari River. Consequently, the lower BIT index values in the delta in the dry season compared to in the wet season are not linked to preferential transport or production of brGDGTs and/or crenarchaeol (Fig. 3e), but related to reduced river discharge during the dry season (Kirkels et al., 2020b). The low discharge results in the intrusion of seawater that naturally contains higher concentrations of crenarchaeol compared to freshwater (Hopmans et al., 2004).

## 5.4 Connecting the Godavari River and the Bay of Bengal


BrGDGTs in continental margin sediments have been widely employed as proxy for fluvially-discharged soil OC, as well as a terrestrial paleothermometer. Both applications are based on the assumption that they represent an integrated signal of the adjacent river system. However, the brGDGT signal that is discharged by the modern Godavari River appears to be biased towards the lower basin. Due to the limited number of studies that include all facets of the land-river-sea continuum, the fate

of fluvially-discharged brGDGTs in the marine realm poorly understood, as well as changes in brGDGT-mineral associations at the fresh/seawater transition.

### 5.4.1 Godavari delta

To evaluate the possible influence of salinity changes on brGDGTs we target riverbed sediments collected along a transect in the main delta branch of the Godavari River. During the dry season, the salinity along this transect increased from ~25 to 29

psu towards the Godavari river mouth (Kirkels et al., 2020b). The brGDGT distributions in the sediments show a decrease in the relative abundances of 6-methyl brGDGTs IIa' and IIIa' compared to their 5-methyl counterparts IIa and IIIa (FIG??). This trend resembles the strong shift from IIa' and IIIa' in the Yenisei River to more IIa and IIIa in the Kara Sea observed by De Jonge et al (2015b). They attributed this change to rapid degradation of the labile, in-river produced OC represented by 6-methyl brGDGTs, resulting in the relative enrichment of the initial (pre-aged) soil-derived brGDGT signal dominated by 5-

methyl brGDGTs. The shift in brGDGT distributions along the salinity transect in the Godavari River delta corresponds with the extent of seawater intrusion (Kirkels et al., 2020b), suggesting that this change may be attributed to either marine brGDGT production and/or a loss of terrestrial/fluvial brGDGTs from mineral surfaces in this transition zone. Indeed, in the Yangtze River estuary both processes were found to occur simultaneously in response to increasing salinity (Cao et al., 2022) and brGDGT production was also linked to the salinity gradient in Svalbard fjords (Dearing Crampton-Flood et al., 2019b). A

potential loss of brGDGTs may be due to active OC processing at the freshwater/saltwater interface as well as to changes in ionic strength, which may reduce the binding of brGDGTs to mineral particles (e.g., Blattmann et al., 2019; Cao et al., 2022; Hou et al., 2020).



In contrast to in the dry season, brGDGT distributions do not change in the wet season, when the river plume extends beyond the Godavari mouth and the water is less saline (<3 psu) (Sridhar et al., 2008; Kirkels et al., 2020b). The seasonal difference in salinity indicates once more that the soil signal is only effectively exported to the Bay of Bengal in the wet season, and supports previous studies that have shown that substantial soil mobilisation is needed (e.g., by a distinct season with increased precipitation) to export soil-derived brGDGTs downstream (Guo et al., 2020; Kirkels et al., 2020a; Märki et al., 2020). Nevertheless, given the reach of the transect we cannot exclude that in the wet season, brGDGT distributions will be still affected by a salinity change further offshore.

### 5.4.2 Bay of Bengal

BrGDGT distributions in a marine sediment core retrieved from the Bay of Bengal in front of the Godavari River mouth and spanning the Early to Late Holocene are distinctively different from those in the modern Godavari basin (Fig. 4a, 6b). Specifically, the Bay of Bengal core has lower relative abundances of 6-methyl brGDGTs, reflected by lower IR values of 0.61-0.66 (Fig. 2k, 5d). The relative abundances of IIIa and IIIa' are higher than in most of the soils, SPM, and riverbed sediments from the Godavari basin (Fig. 4a). As a result, the Bay of Bengal core sediments are clearly separated from all other samples when they are passively added to the PCA of the brGDGTs in SPM and riverbed sediments from the modern Godavari basin (Fig. 6b). The distinct brGDGT distribution in the Bay of Bengal sediments from that in the modern Godavari basin points towards a different source of the brGDGTs. This implies that the majority of the soil-derived brGDGTs is not effectively transferred to the sedimentary archive and/or overprinted by marine in situ production during transport through the water column, or after deposition on the sea floor. Low BIT index values in the Bay of Bengal core sediments (≤0.07; Fig. 2j, 5c) indicate that most of the OC in these sediments indeed has a marine origin. The limited contribution of terrestrial brGDGTs to the core location fits the observations of a rapid reduction in fluvially discharged brGDGTs offshore at e.g., the Siberian (Sparkes et al., 2015), Iberian (Zell et al., 2015; Warden et al., 2016) and East China continental shelf (Zhu et al., 2011).

The occurrence of marine in situ production of brGDGTs is increasingly recognised on continental margins, where it can be recognised based on a high degree of cyclisation of the tetramethylated brGDGTs (e.g., Peterse et al., 2009; Zhu et al., 2011; Zell et al., 2015; Sinninghe Damsté, 2016). Sinninghe Damsté (2016) defined #rings$_{tetra}$ >0.7 as a cut-off value to identify a predominantly marine source of brGDGTs. However, the #rings$_{tetra}$ in the Bay of Bengal core is relatively low (~0.29; Fig. 5e) and falls within the range covered by soils (0.36±0.02), SPM (dry season: 0.37±0.02; wet: 0.25±0.01) and riverbed sediments (dry season: 0.41±0.02; wet: 0.40±0.03) in the modern Godavari basin. The increase in #rings$_{tetra}$ in the marine realm is proposed to be a response to the relatively more alkaline conditions in the marine environment compared to those in soils (Sinninghe Damsté, 2016). Such a contrast is absent in the Godavari River system, where the pH of soils (~8) and seawater in the Bay of Bengal (~8.1) is similar (Sarma et al., 2015). Moreover, Sinninghe Damsté (2016) found that most brGDGT production takes place in the zone between 50-300 m water depth, whereas #rings$_{tetra}$ appears to decrease again at greater water depths (Zell et al., 2015; Sinninghe Damsté, 2016). The latter is in agreement with the low #rings$_{tetra}$ values at the deeper location of the Bay of Bengal core (~1250 m water depth). Indeed, recent studies investigating deep marine trenches (1.6-11





km), which hardly receive any terrestrial input, show a distinct brGDGT distribution from shallower marine sediments, and have lower values for #rings$_{tetra}$ and IR values (Xiao et al., 2020; Xu et al., 2020).

To further decipher the provenance of brGDGTs in the marine environment, Xiao et al. (2016) proposed that a ratio of (IIIa + IIIa')/(IIa + IIa') can be used. This ratio differentiates brGDGTs derived from soils, where this ratio is <0.6, those produced in

continental shelf sediments (≤5000 m water depth), where this ratio is mostly >1, and in deep (>11000 m) marine settings, where this ratio is >5 (Xiao et al., 2016, 2020; Xu et al., 2020). In the Bay of Bengal core, values for this ratio are mostly >1 (0.77-1.34, 1.07±0.02; Fig. 5f) and are significantly higher than in soils (<0.20) and SPM and riverbed sediments (<0.33) from the modern Godavari basin in both seasons. This indicates that despite the low #rings$_{tetra}$, the brGDGTs in the Bay of Bengal core do likely have a marine origin. Indeed, samples from the modern Godavari basin and the marine sediment core are clearly

separated in a crossplot of the (IIIa + IIIa')/(IIa + IIa') ratio versus the contribution of IIIa' (Fig. 9), and closely match the distinct trends of global soils and coastal/shelf sediments.

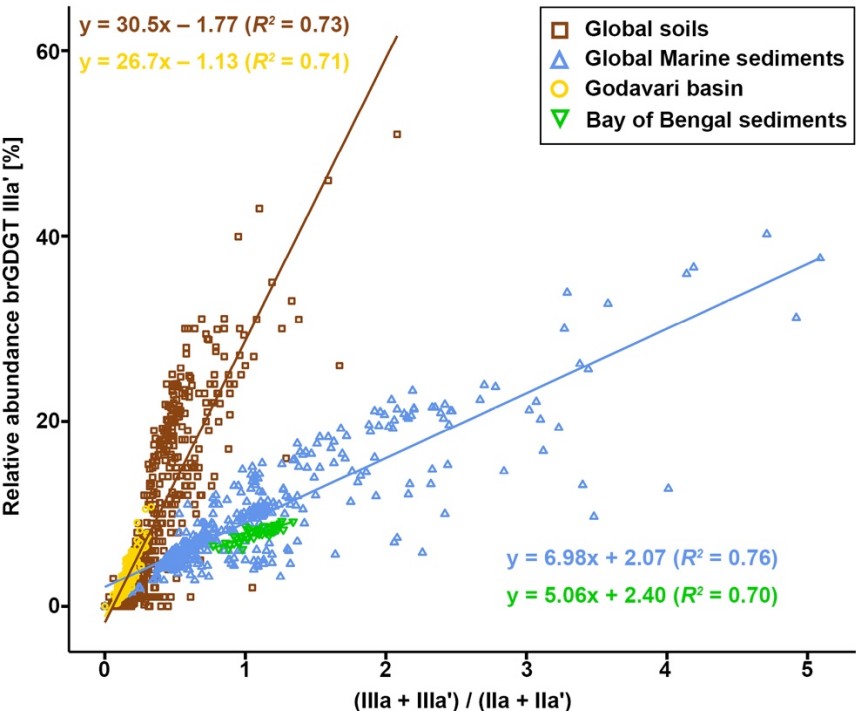

**Figure 9 – Crossplot of the ratio (IIIa + IIIa') / (IIa + IIa') versus the relative abundance of IIIa' for globally distributed soils (square)**
**and marine sediments (triangle) after Xiao et al. (2020) and Godavari basin samples, including soils, SPM and riverbed sediments collected the dry and wet seasons (circle) and the Bay of Bengal Holocene sediments (inverted triangle). The solid lines denote the linear fit, and the correlations and R² are shown.**



### 5.4.3 Soil to sea continuum

The apparent absence of terrestrially-derived brGDGTs in the Holocene Bay of Bengal sediment core contrasts with the presence of n-alkanoic acids derived from plant-wax lipids in the same sediments (Ponton et al., 2012; Usman et al., 2018). This discrepancy points towards different intrinsic stabilities, mineral-binding, or transport mechanisms for these specific biomarkers. Similarly, Hou et al. (2020) reported that the burial efficiency of plant-derived n-alkanes differed from that of n-alkanoic acids in continental margin sediments in the South China Sea, revealing that these terrestrial biomarker lipids have a

distinct fate. In addition, different lipid biomarkers may derive from different parts of the river basin, as has been shown in the Danube (Freymond et al., 2018b) and Congo River basins (Hemingway et al., 2016), possibly linked to differences in their degree of binding in mineral-associations and/or mode of transport (Eglinton et al., 2021).

In contrast to the modern Godavari basin, where sediment and brGDGT export is biased towards the lower basin, the sediments in the upper part of the core that represents the Late Holocene are smectite-rich and originate from the Deccan basalts region

in the upper Godavari basin (Giosan et al., 2017; Usman et al., 2018). However, smectite may be partially stripped of associated soil OC following changes in ionic composition and strength at the mouth-to-margin transition, where the mineral surfaces may be re-occupied by marine-produced OC (Blattmann et al., 2019). Since brGDGTs are presumably slightly less hydrophobic than plant wax n-alkanoic acids, their association with mineral particles may be weaker. This could result in preferential loss of brGDGTs from mineral particles over n-alkanoic acids upon entering the marine realm, and lead to

differential settling, enrichment, and preservation of certain biomarkers in sedimentary archives (Blattmann et al., 2019).

When brGDGT concentrations are normalised to mineral surface area (MSA, see Usman et al., 2018), they are 1 to 4 order(s) of magnitude lower than those of n-alkanoic acids in the same set of Godavari soils, dry season riverbed sediments, and Holocene Bay of Bengal sediments. While n-alkanoic acid loadings remain in the same range along the soil-to-sea continuum (Usman et al., 2018), brGDGT loadings decrease on average by a factor ~3 from the Godavari basin to the Bay of Bengal core

(Appendix 3). This discrepancy indicates that these compounds have a different transfer/burial efficiency, and would imply that the choice of biomarker that is used as specific tracer may influence estimates of the (long-term) sink capacity of marine sediments in the global C cycle, as well as their comparability in multi-proxy paleoreconstructions.

### 5.5 Implications for brGDGT-based paleoreconstructions

The inferred predominantly marine origin of brGDGTs in the Bay of Bengal sediments hampers their use as paleothermometer

for the nearby river catchment. Indeed, brGDGTs in the Bay of Bengal sediments translate into continental mean annual air temperatures (MAATs) between ~20 and 22°C (Fig. 5h), using the MBT'$_{5me}$ index and the BayMBT$_0$ model (Dearing Crampton-Flood et al., 2020). These reconstructed MAATs are lower than the basin-average, measured MAAT of ~27°C for the modern Godavari basin. Methods to correct for marine contributions to soil-derived brGDGTs based on the offset in #rings$_{tetra}$ between marine sediments and catchments soils (Dearing Crampton-Flood et al., 2018) are not applicable to the

Bay of Bengal core due to the water depth >1000 m of the site and the similar #rings$_{tetra}$ in the Bay of Bengal core and the





modern Godavari basin. Alternatively, the ratio proposed by Xiao et al. (2016, 2020) may be further explored to quantitatively assess the contribution of marine in situ production to the brGDGT pool in the core, provided that studies across environmental gradients confirm robust terrestrial, freshwater and marine end-members.

Interestingly, from ~3000 yr. BP there is a small change in the composition of brGDGTs in the core sediments, with slightly

higher BIT, IR and MBT'$_{5me}$ values and a lower relative abundance of brGDGT Ia, as well as #rings$_{tetra}$ and (IIIa + IIIa')/(IIa + IIa') ratio values (Fig. 5b-g). This change corresponds with a steep increase in the sediment flux from the upper Godavari basin to the Bay of Bengal that was inferred from a change in detrital neodymium isotope values and the stable carbon isotopic composition of fatty acids (Giosan et al., 2017; Usman et al., 2018). This increase in soil erosion was attributed to natural causes such as aridification and a reduced vegetation cover and anthropogenic changes that resulted in increased agricultural

land use and population expansion in the Godavari basin. The minor shift in brGDGT distributions could speculatively be linked to changes in the ratio of terrestrial-to-marine input. Alternatively, the shift may be explained by a change in the marine brGDGT-producing community, possibly in response to these changing inputs from the river system. Interestingly, isoprenoid GDGTs that are produced by marine archaea and are preserved in the same core do not show this change in composition at ~3000 yr. BP (Fig. 5h). Instead, they show that the sea surface temperatures (SST) consistently remained between ~29 and

30°C over the Holocene (Fig. 5h), similar to estimates based on Mg/Ca records in the Western Bay of Bengal during the same period (Govil and Naidu, 2011). This observation would suggest that brGDGTs with a marine origin respond to different environmental forcing than isoprenoid GDGTs and/or are produced in a distinct part of the system, i.e., in sediments or the lower water column, instead of in (sub)surface waters.

Taken together, our study provides a clear example of the challenges when using brGDGTs as tracer for soil OC along the

land-river-sea continuum, or as paleothermometer in marine sediments. Our findings support the statements of Dearing Crampton-Flood et al. (2021), who demonstrated the importance of determining the exact source(s) of brGDGTs prior to using them in temperature reconstructions, and highlights the need for more advanced methods to disentangle a mixed soil, riverine and marine in situ produced brGDGT distribution in sedimentary settings.

## 6 Conclusions

Our comprehensive study of brGDGTs in soils, SPM and riverbed sediments in the modern Godavari River basin allows us to trace soil OC along the entire soil-river-sea continuum. The hydrological contrasts between the dry and wet season, and spatially between the upper and lower basin linked to the natural change in bedrock geology, provide insight in the timing and source(s) of brGDGT and sediment transport through the Godavari River system. High contributions of 6-methyl brGDGTs in SPM and riverbed sediments indicate that brGDGTs are produced in situ in the river in the upper basin year-round, and

throughout the entire basin in the dry season. Soil OC is only substantially mobilised and exported by the Godavari River to the Bay of Bengal in the wet season. The exported brGDGT distributions is biased towards the areas that receive most





precipitation in the lower part of the Godavari basin. This trend is amplified by the presence of dams that limit fluvial transport from the upper basin.

Comparison of the OC content, brGDGT concentrations and distributions in soils and sediments with the elemental

composition of the same soils and sediments indicates that bulk OC is transported in association with minerals, while brGDGTs may be less tightly bound to mineral particles. Regardless, sediments and brGDGTs share a common origin in the lower Godavari basin. However, this coupling may also result from the turbid conditions that hamper in situ production, in combination with rapid transport downriver linked to the high precipitation in the lower basin. River depth profiles and sediment size fractions reveal no hydrodynamic sorting effects on brGDGT distributions. Nevertheless, the fluvially discharged

brGDGT distributions does not match that of brGDGTs in a Holocene sediment core retrieved in front of the Godavari mouth in the Bay of Bengal. This offset indicates that the soil-derived signal is rapidly lost after fluvial discharge and/or overprinted by marine in situ production. The observed disconnection between the Godavari basin and the Bay of Bengal sediments would impact brGDGT-based estimates of the capacity of marine sediments as long-term sink for soil OC, and also limit the use of brGDGTs in marine sediments for terrestrial paleoreconstructions.



**Appendix 1**

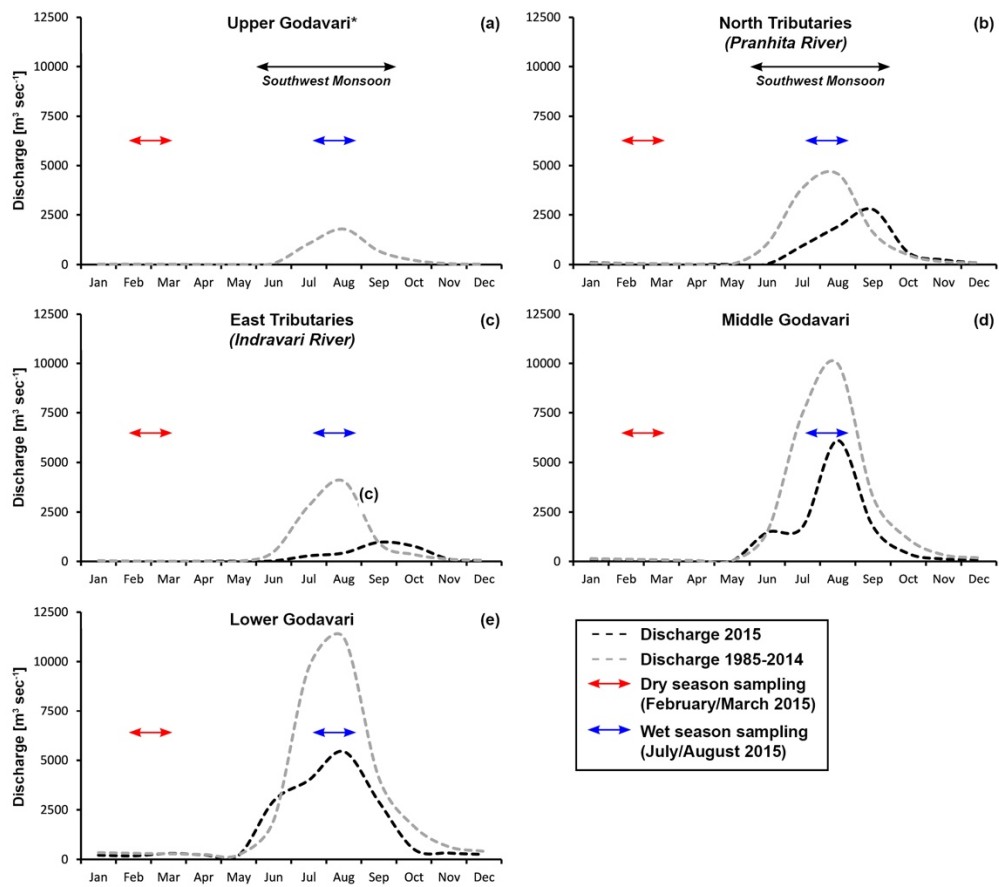

Monthly water discharge data for 2015 and the long-term average for 1985-2014 (India Water Resources Information System, accessed 01/07/2021) for the Godavari subbasins, (a) Upper Godavari (main stem, station at Mancherial, near site 31; * no data are available for April-December 2015), (b) North Tributaries (Pranhita River, station at Tekra, near site 32), (c) East
Tributaries (Indravati River, station at Pathagudem, near site 26), (d) Middle Godavari (main stem, station at Perur, ~100 km upstream of site 28), (e) Lower Godavari (main stem, station at Polavaram, near site 13 (inflow of Reservoir Lake at Rajahmundry)). The arrows indicate the period of sample collection in the dry season (red; February/March 2015) and in the wet season (blue; July/August 2015). The Southwest Monsoon (SWM) brings the majority of rainfall (75-85%) in the period June-September. For rainfall distributions across the Godavari basin and variability in Southwest Monsoon precipitation, see
Kirkels et al. (2020b).



**Appendix 2**

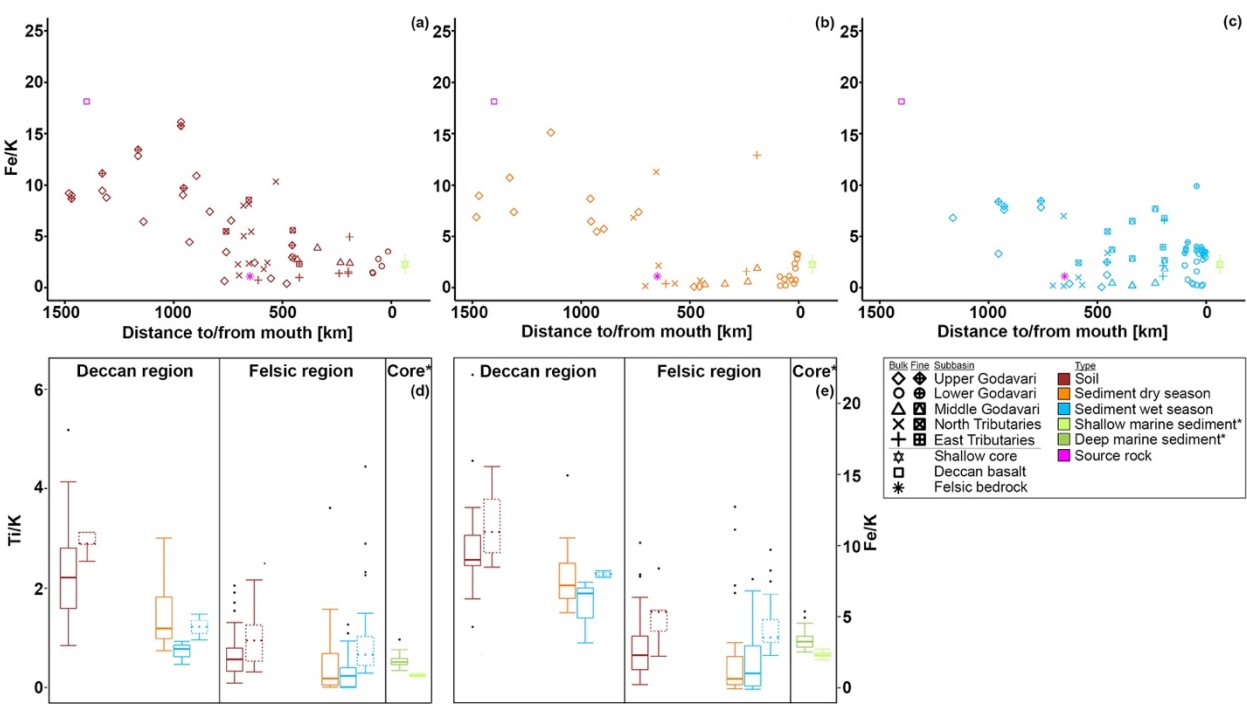

Fe/K ratios in (a) soils and riverbed sediments collected in the (b) dry and (c) wet season versus distance to/from the river mouth. The square and star (magenta) represent the end-member values for the Deccan basalts (Das and Krishnaswami, 2007)

and felsic bedrocks (Archean Proterozoic Gneiss Complexes, APGC) (Moyen et al., 2003), respectively. The outlined star (greenyellow) represents the average of marine sediments retrieved from shallow cores in front of the Godavari mouth (0-48 cm below sea floor, ~300 yr.; Kalesha et al., 1980) (error bar represents the standard deviation: ±0.83). Box-and-whisker plots of (d) Ti/K and (e) Fe/K ratios for the Godavari basin soils and sediments collected in the Deccan basalts and felsic regions and in marine sediments in the Bay of Bengal. * Core data are retrieved from shallow (0-48 cm below sea floor, ~300 yr.;

Kalesha et al., 1980) and deep marine sediments (0-300 m (NGHP-1-3B) and 0-184 m bsf (NGHP-1-5C), no age model used; Mazumdar et al., 2015). The box represents the first (Q1) and third (Q3) quartiles, and the line in the box represents the median value, the whiskers extent to 1.5×(Q3-Q1) values and outliers are shown as points. Solid lines represent bulk data, dashed lines the fine fractions (≤63 μm).





**Appendix 3**

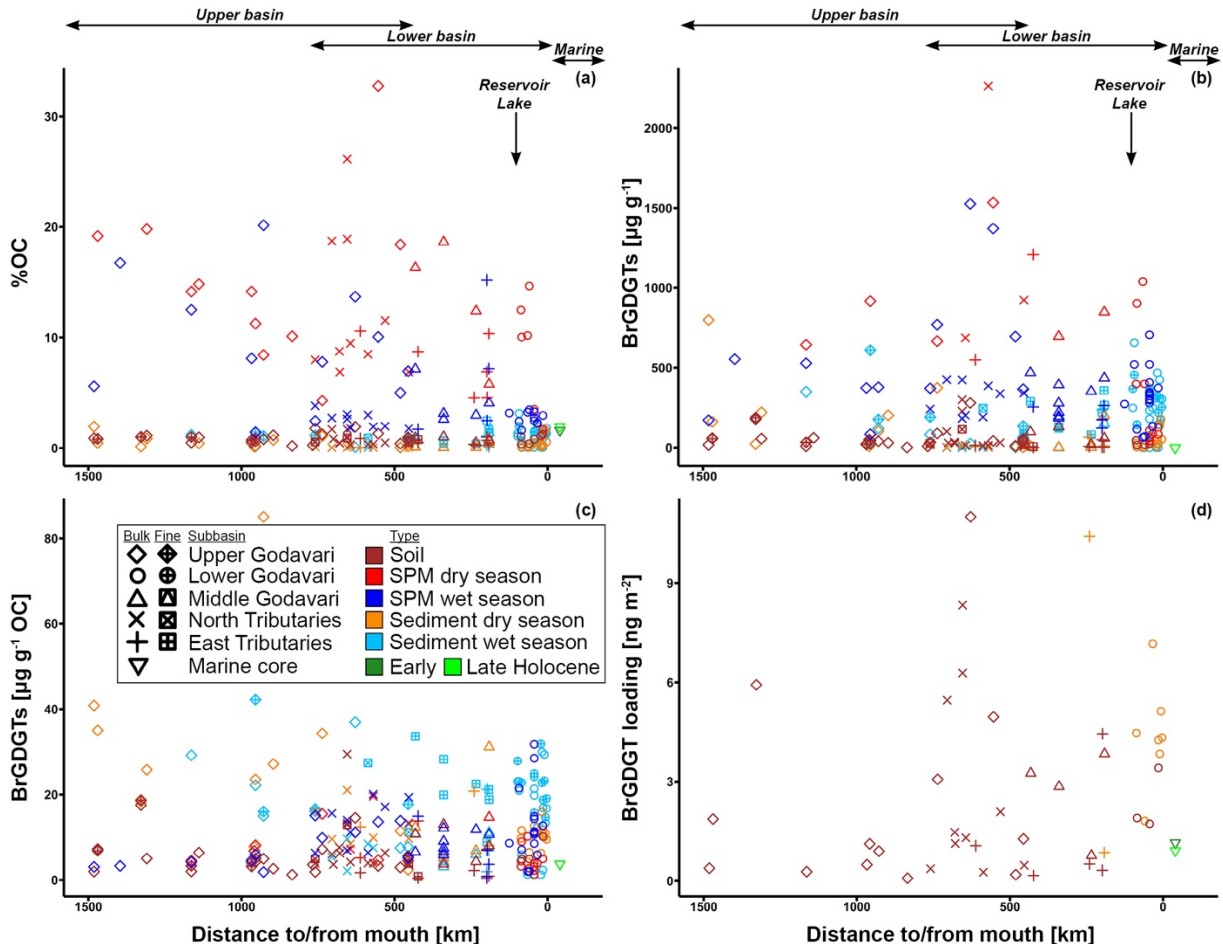


Evolution of %OC and brGDGTs along the Godavari River. (a) %OC, (b) brGDGT concentration normalised to sediment weight, (c) brGDGT concentration normalised to OC versus distance to/from the river mouth for soils, SPM and riverbed sediments and average values for Holocene sediments in the marine realm (error bars (±SE) are too small to be visible), and (d) brGDGT loadings, i.e., normalised to Mineral Surface Area (MSA, see Usman et al., 2018) for Godavari soils (n=46),

riverbed sediments collected in the dry season (n=9) and average values for Holocene sediments in the marine realm (n=46; error bars (±SE) are too small to be visible). The upper basin is defined as the Upper Godavari and two sites (42 and 43, at 655 and 759 km from the mouth, respectively) in the North Tributaries. The lower basin comprises the other North Tributary sites, the East Tributaries, Middle and Lower Godavari. The vertical arrow represents the location of the Reservoir Dam at Rajahmundry. The symbols refer to the different subbasins, and the different fractions (bulk and fine fractions ≤63 μm). The

colours refer to the different sample types.
**Data availability**

Geochemical and GDGT data associated with this article are available in the Pangaea Data Repository (Kirkels et al., 2021a) and can be accessed at https://doi.org/10.1594/PANGAEA.934712. GDGT data for core NGHP-01-16 associated with this article are available in the Pangaea Data Repository (Kirkels et al., 2021b) and can be accessed at
https://doi.org/10.1594/PANGAEA.934701. Branched GDGT data of the bulk soils are also accessible in Pangaea (Dearing Crampton-Flood et al., 2019a) at https://doi.org/10.1594/PANGAEA.907818.

**Author contribution**

FMSAK, TIE, and FP designed the study; FMSAK, HMZ, MOU, and FP planned and carried out fieldwork; CP and LG provided material from the Bay of Bengal core; FMSAK, HMZ, MOU, and CP carried out laboratory analyses; FMSAK, 925   HMZ, SH, and FP interpreted the data; FMSAK took lead in preparing the manuscript with input from all co-authors.

**Competing interests**

The authors declare that they have no conflict of interest.

**Acknowledgements**

This work was supported by the Netherlands Organisation for Scientific Research (NWO) [Veni grant 863.13.016 to FP]. The
fieldwork for this study was carried out with help from Prof. P. Sanyal from the Indian Institute of Science Education and Research in Kolkata (India). We thank dr. M. Lupker, C. Martes and dr. S. Basu for their help with the pre-monsoon fieldwork campaign. For technical support in the laboratory, we thank D. Kasjaniuk, A. van Leeuwen-Tolboom, dr. K. Nierop, J. van Tongeren (general support and brGDGT analysis) and C. Mulder (HF fumigation and ICP-MS) (Utrecht University, The Netherlands) and dr. E. Hopmans and dr. M. van der Meer (brGDGT and TOC analysis) (NIOZ, The Netherlands).

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
