# Peer review of "From soil to sea: Sources and transport of organic carbon traced by tetraether lipids in the monsoonal Godavari River, India"

_Biogeosciences, 2022_

## Author Response (AR1)

Utrecht, 9 August 2022

Dear editor, dear Sebastian Naeher,

Thank you for your positive evaluation of our manuscript and our replies to the comments of the reviewers.

We have revised the manuscript according to the changes we proposed in our replies to the reviewers. Specifically, we have 1) shortened the results section where possible, although 2) we have added isoGDGT data. Note that we have limited the discussion of the isoGDGT data to their use as independent support for our interpretations of the brGDGT data that remain the main focus of our work. In short, the distinct f(cren') and the concentration of isoGDGTs for soils and river-derived material (SPM and riverbed sediments) indicates that isoGDGTs have a mostly aquatic source in the Godavari River, and, therefore, cannot aid in tracing soil OC through the river basin. Regardless, elevated values of GDGT-0/crenarchaeol in SPM and riverbed sediments from the upper basin, where contributions of 6-methyl brGDGTs are high year-round, suggest that aquatic brGDGT production is likely favored by low oxygen conditions.

We have decided to keep the MBT'$_{5Me}$ as part of our manuscript, as it helps to visualize changes in the relative distribution of brGDGTs between the distinct sample types/compartments of the river basin. Furthermore, we believe that it is very clear from our discussion (section 5.5 Implications for brGDGT-based paleoreconstructions) that brGDGTs can only be used as proxies if certain criteria are fulfilled (based on the BIT index, the #rings$_{tetra}$, IR), and that this is not the case at our site.

In addition to the requested changes, we have adjusted the maps with precipitation data in Figs. 1 and 7 to make them colorblind friendly as well as for consistency with our manuscript on vegetation change in the Godavari Basin (accepted for publication in Biogeosciences pending minor changes: https://bg.copernicus.org/preprints/bg-2022-57/).

A point-to-point list of changes can be found below. All changes in the manuscript are made with track changes on.

We hope that you find this revised manuscript suitable for publication in Biogeosciences.

On behalf of all co-authors,
Francien Peterse

**Reviewer #1, Dr. Naafs**

- I suggest you shorten the results section. It is very long with a lot of details that sometimes make it hard to follow and some details appear not to be necessary. Condensing the results by focussing on the key results will improve the readability of the manuscript.

*Reply: We agree that the results section is very 'complete' and will shorten this section to include only the information that is directly relevant to the interpretation of our data in a revised manuscript.*

**Changes made: We have done this.**

-Why are the isoGDGTs not discussed? Crenarcheaol is used for BIT, but what about the others? For example cren/cren' ratios can tell us something about the potential source organisms and this differs between mineral soils and aquatic production in some places (Li et al., 2016). The isoGDGTs are measured already (I assume) as part of the brGDGT runs, so potentially there is a lot of extra information available with minimal effort?

*Reply: Thank you for the suggestion. We do indeed have isoGDGT data available. We did not include any of this data in the initial manuscript because we thought that the brGDGT dataset as such was already extensive enough, containing soils, SPM from both wet and dry seasons, riverbed sediments from both wet and dry seasons, as well as fine (<63um) fractions, and a marine sediment core. We were afraid that the manuscript would become too dense and long, and would loose its focus if the isoGDGTs would be added. In addition, the isoGDGT dataset for the Godavari basin is part of a manuscript in preparation by Martinez-Soza et al.*

*Nevertheless, if the reviewers and editor think that the isoGDGTs are a valuable addition to the current work, we will of course consider this in a revised manuscript. A quick first analysis reveals that the isoGDGT data lead to similar conclusions as the brGDGTs:*

- *$f(cren')$ is slightly higher in soils (on average 0.15) than in SPM, riverbed sediments and the marine core (average range 0.05-0.10), indeed implying different producers in soils and aquatic environments as suggested by Dr. Naafs.*
- *GDGT-0/cren is higher in SPM and riverbed sediments collected during the dry season than in the wet season (on average 1.6 vs 0.9), likely due to anoxic conditions in stagnant waters during the dry season and in the upper basin, facilitating methanogens that contribute to GDGT-0.*
- *GDGT-2/GDGT-3 is higher in the marine sediment core (on average 3.8) compared to soils, SPM, and riverbed sediments in the river basin (1.0-1.2), indicating that the isoGDGT signal that is discharged by the river is overprinted by isoGDGTs produced by marine Thaumarchaeota. Similarly, GDGT-0/cren is substantially*

*lower in the marine sediment core (0.2) than in the river basin, and represents 'normal' marine conditions. This is also confirmed by the low BIT index values (around 0.05) reported in Fig. 5c of the original submission.*

**Changes made: We have included the OC-normalized concentration of isoGDGTs, f(cren') and GDGT-0/crenarchaeol values for all sample types to the results section. Based on the concentration and relative abundance of the isoGDGTs, it is clear that these compounds have an aquatic source in SPM and riverbed sediments and are thus not suitable as tracers for soil OC (i.e., the main focus of this work). Therefore, we have limited the discussion on isoGDGTs and only use them as independent support of our interpretation of the brGDGT data where appropriate (mainly section 5.2 on the sources of GDGTs in the Godavari River).**

-Why is the focus on core GDGTs and not IPLs? For the SPM samples especially, would it not make sense to look at the IPLs to determine in situ production? The signal in the IPLs might be even stronger compared to the core GDGTs?

*Reply: We agree that IPLs could have provided a stronger argument for in situ production in the river. However, the logistics in the field did not allow storing our samples frozen after sampling and transport to the lab. Only the SPM filters were stored at 4 °C, but soils and sediments were kept at ambient temperatures during our 1 month field expeditions. Since IPL headgroups are considered to be quickly lost upon cell death, we anticipated that the remaining IPLs (if any) in the samples would not be a reliable representation of the initial IPL abundance, and thus decided to focus on core GDGTs instead.*

**Changes made: we have added a few lines to the methods (section 3.5) to explain that we assume that the majority of the IPLs will be degraded to CLs due to sample storage at ambient temperature during our expedition and transport back to the lab, in combination with using the ASE for biomarker extractions under high temperature and pressure.**

Related to this, I see (line 221) that some fractions were saponified, but others were not. Although not extracted with a BD protocol, this saponification of the TLE might release IPLs. This affects what fraction of the GDGTs you look at (core for the non-saponified and a mixture of IPL-derived cores and cores for the saponified samples). Couldn't this difference in sample work up in theory explain some of the observed differences between the different sample types? This needs more explanation.

*Reply: We understand the concern of the reviewer, but we believe that the differences in GDGT concentrations will be marginal. Firstly, samples have been stored non-frozen during fieldwork and transport, facilitating the degradation of IPLs on the road. And secondly, the extraction with the ASE uses high temperature and pressure, which will also degrade IPLs into core lipids during the process. We, therefore, believe that saponification of the obtained TLE will not further release substantial amounts of IPL brGDGTs.*

*In addition, the %IPL-derived brGDGTs in soils is generally much lower than the pool of 'fossil' brGDGTs that are present in the soil as core lipids (e.g. Peterse et al., 2010; Huguet et al., 2010; Zell et al., 2013), which thus represent the majority of the brGDGT signal. This is also true in river SPM (e.g., Zell et al., 2013; De Jonge et al., 2014). Given that IPL-derived brGDGTs and core lipid brGDGTs generally have a similar distribution in soils and river SPM, the work up procedure followed here is not considered to introduce large differences in brGDGT distributions nor concentrations the dataset.*

*We will briefly clarify our assumptions in a revised version.*

**Changes made: we have also addressed this issue in the lines that we added to the methods (section 3.5).**

Other minor comments and typos:

Lines 64-66: both papers cited here are using mineral soils, not peat.

*Reply: We will add a reference to Naafs et al., 2017 to also cover peats.*

**Changes made: we have done this.**

Line 75: also cite culture results from (Halamka et al., 2021)

*Reply: We will add this reference.*

**Changes made: we have done this.**

Line 97: is this due the overall higher pH in rivers compared to soils?

*Reply: this is indeed the mechanism that has been proposed by De Jonge et al., 2014. We will clarify this in the revised version.*

**Changes made: we have done this.**

Results: I suggest you shorten the results section. It is very long with a lot of details that sometimes make it hard to follow. Condensing the results through focussing on the key results will improve the readability of the manuscript.

*Reply: As mentioned earlier, we agree with the reviewer here and will revise the results section to improve the readability.*

**Changes made: we have done this.**

Figure 5 (and associated text); In samples with such a low BIT values, can we ever use MBT'5me? Not sure it makes sense to show this data in this graph.

*Reply: The application of MBT'5me in marine settings should always be done with caution and after a thorough assessment of the source(s) of brGDGTs, like we suggest in discussion section 5.5 and the conclusion. Note that a low BIT index does not necessarily indicate little terrestrial input; after all, the BIT index is a ratio and terrestrial input can be masked by enhanced marine production. For example, BIT index values were low in Pliocene North Sea sediments, whereas d13C of the organic matter indicated a primarily terrestrial origin of this material and brGDGTs could be used to infer paleotemperatures for the nearby land (Dearing Crampton-Flood et al., 2018).*

*In our manuscript, we decided to include MBT'5me in this part of the discussion for the completeness and to enable the comparison with MBT'5me values for the Godavari basin. Importantly, after an assessment of the sources of the brGDGTs -as we suggest to always do before using it as paleothermometer- we do not interpret the MBT'5me record as paleotemperatures. Regardless, we can reconsider including MBT'5me in the revised version if the editor advises us to do so.*

**Changes made: we have kept the MBT'5Me as part of our manuscript and clearly state that brGDGTs in marine sedimentary archives can only be used if certain criteria are fulfilled, which, in the Bay of Bengal, is not the case.**

Line 485: Figure 5?

*Reply: the brGDGT distributions are also given in Fig. 4a, but we will add Fig. 5 here too.*

**Changes made: we have done this.**

Line 510: how does this fit the with brGMGT data (Kirkels et al., 2022)?

*Reply: Kirkels et al., 2022 report that brGMGTs are not widespread in the Godavari basin, in contrast to the marine sediment core, where brGMGTs are continuously present. The occurrence of brGMGTs in the basin appears to be determined by low oxygen/high nutrient conditions (e.g. agricultural soils, inundates soils, stagnant waters) rather than soil type and can, therefore, also not be used to trace basin-specific contributions.*

**Changes made: none.**

Line 525: you mean low BIT?

*Reply: We are not sure what the confusion is here, as this line already says low BIT.*

**Changes made: none.**

Line 539: PCA

*Reply: We assume that the reviewer refers to the second half of this line where we write "This PC further...". PC refers to PC2 in the previous sentence, not to the PCA. We will clarify this.*

**Changes made: we have clarified this.**

Line 573: Cite (Halamka et al., 2021)

*Reply: we will add this citation where relevant.*

**Changes made: we have done this.**

Line 598-604: Explore broader isoGDGT distribution to provide more insights into the archaeal source, for example cren/cren' ratios, etc.

*Reply: see our reply to the earlier comment of this reviewer.*

**Changes made: we have done this, see our earlier replies.**

Line 741: FIG??

*Reply: Thank you for spotting this! In the end we decided to not add yet another figure to support this statement, but instead just describe it in the text. We will remove this reference.*

**Changes made: we have done this.**

**Reviewer #2**

To make this paper a more attractive read, I suggest to change the title and headers of the discussion from "descriptions" to actual statements that reflect the main findings. For instance, the title currently does not really reflect the main finding that soil brGDGT signals are overprinted by riverine and marine in situ production.

*Reply: Thank you for the suggestion. We will change the titles and headers of the discussion to match the content/main finding of each section.*

**Changes made: we have done this where appropriate.**

Ln 67: We now know of several bacteria that synthesize brGDGTS (Halamka et al., 2021 doi: 10.7185/geochemlet.2132; Sinninghe Damsté et al., 2018). Therefore, I wouldn't doubt that bacteria are truly their source and consequently would use another word than enigmatic.

*Reply: We agree with this comment. The enigmatic was more directed towards the exact bacteria that may produce brGDGTs. We will rephrase this sentence.*

**Changes made: we have done this.**

Ln 70-73: There is recent evidence that there are also bacteria that do not produce iso-diabolic acids that synthesize brGDGTs (Halamka et al., 2022 https://doi.org/10.31223/X5WD2C), therefore, I suggest to be more careful with the statements made here.

*Reply: On hindsight, we decided to constrain our references to work that has passed the scientific peer-review procedure. We will revise this sentence to match the latest findings, but will refrain from referring to this work, as well as that of Chen et al., 2022 in the revised version.*

**Changes made: we have done this.**

Ln 75: Please also acknowledge the work of Halamaka et al. (2021) here.

*Reply: A similar request was made by the other reviewer Dr Naafs. We will add this reference.*

**Changes made: we have done this.**

Ln220 onwards: Does this mean that these samples (dry season SPM, riverbed sediments, and fine fractions of soils) were not saponified, while wet season SPM and bulk soils were saponified? Why were these samples treated differently? Saponification may release also some IPL-GDGTs as core GDGTs and affect ratios, also of isoGDGTs to brGDGTs. Have the authors considered the effect of this? Also, there is no reference for the Al2O3 column separation, was this tested for the effectiveness (and yields) for core GDGTs?

*Reply: A similar comment was made by the other reviewer. Indeed, only wet season SPM and soils were saponified to isolate fatty acids used in the study by Usman et al., 2018. There, the choice was made to only study material collected during the wet season when most soil mobilization and transport is taking place. Since the isolation of fatty acids requires additional steps in the workup procedure and we had a large number of samples (>300), we decided to optimize the workup procedure for our target compounds brGDGTs.*

*In addition, we believe that the potential contribution of IPL-derived brGDGTs to the measured brGDGT signal will be marginal due to the following reasons:*
*- Logistics in the field did not allow us to store our samples frozen after sampling and transport, facilitating the degradation of IPLs on the road.*
*- Our samples have been extracted with the ASE that uses high temperature and*

*pressure, which degrades IPLs in the process.*
*- The %IPL-derived brGDGTs in soils is generally much lower than the pool of 'fossil' brGDGTs that are present as core lipids (e.g. Peterse et al., 2010; Huguet et al., 2010; Zell et al., 2013), which thus represent the majority (>80%) of the brGDGT signal. This is also true in river SPM (e.g., Zell et al., 2013; De Jonge et al., 2014). Given that IPL-derived brGDGTs and core lipid brGDGTs generally have a similar distribution in soils and river SPM, the work up procedure followed here is not considered to introduce large differences in brGDGT distributions nor concentrations the dataset.*

*The separation of total lipid extracts over a $Al_2O_3$ column to isolate a GDGT fraction is a common procedure followed by many labs globally and does not have an original citation.*

***Changes made: we have added a few lines to the methods section to explain the assumptions that we have made, as listed in our reply to reviewer #1.***

Ln 239: Change to APCI

*Reply: we will change this,*

***Changes made: we have done this.***

I find many of the titles in the discussion bland. To keep the reader excited I suggest to instead mention the main finding in the title. For instance instead of "Spatial variations in GDGTs in Godavari soils" you could say "The effect of moisture and temperature on the spatial distribution of in GDGTs in Godavari soils" or instead of "Sources of GDGTs in the Godavari River" you could say "6-methyl-brGDGTs indicate in situ production in the Godavari River"

*Reply: Thank you for this suggestion. We will definitely follow up on this in a revised manuscript.*

***Changes made: we have done this.***

Ln 539: replace "tears" with "teases"

*Reply: We will change this.*

***Changes made: we have done this.***

Ln 550: Please indicate that you are now also referring to Fig. 6a and not only 6b.

*Reply: We will add a reference to the appropriate figure(s) here.*

***Changes made: we have done this.***

Ln 573: Please also give credit here to the paper by Halamka et al., 2021 (doi: 10.7185/geochemlet.2132 )

*Reply: We will add this.*

**Changes made: we have done this.**

Ln576: How was it shown that the brGDGT producing bacteria were heterotrophic?

*Reply: This was based on the d13C value of the hydrocarbons that were released from brGDGTs after ether cleavage in, for example soils that were exposed to labeled $CO_2$ (Weijers et al., 2010), or in soils along a transect away from a natural $CO_2$ vent with a distinct isotopic composition (Oppermann et al., 2010). These studies found that the d13C value of brGDGT-derived hydrocarbons matched that of $CO_2$ in a way that would fit with a heterotrophic lifestyle of their producers.*

**Changes made: none.**

Ln 600: Did the authors see higher absolute amounts of crenarchaeol to confirm a higher activity of ammonia oxidizing archaea?

*Reply: We are not entirely sure what the reviewer would like to know. The high(er) crenarchaeol concentrations in the dry season that is referred to here were reported in a study on the Lower Amazon by Zell et al. (2013). They found that seasonal variations in the BIT index were mostly driven by the production of crenarchaeol in the river. In the Godavari River, crenarchaeol concentrations are (somewhat) higher during the wet season that during the dry season. But more importantly, and in contrast to in the Lower Amazon, the in situ production of brGDGTs are more important in determining the BIT index here than crenarchaeol, as we state in line 602-604.*

**Changes made: none.**

Fig. 7: Can you indicate in this plot again where the border of the Lower and Upper Godavari Basin is and where the North and East Tributary regions are? There is a red dashed line, I assume this is supposed to separate the two basins?

*Reply: The red dashed line indeed separates the Upper and the Lower basin. We will better indicate the different subbasins in a revised figure.*

**Changes made: we have done this.**

5.3 and 5.4 onwards: Again, I recommend to choose more meaningful titles so the reader is informed on the most important points. Suggestions are "5.3.2 Low mineral associations during river transport" "5.3.3 The marine sedimentary brGDGT

composition reflects the lower Godavari basin" or "5.3.4 Absence of size-related sorting in the Godavari River"

*Reply: Again, thank you for the suggestion. We will follow up on this.*

**Changes made: we have done this.**

Ln 710: Do the authors have any idea why the depth profiles of the Godavari River look different to other monsoonal rivers?

*Reply: The relatively little variation in the depth profiles from the Godavari River may possibly be explained by the lower flow velocity of the Godavari compared to that of other monsoonal rivers, especially those with a larger elevation gradient, such as the Amazon River and the Ganges-Brahmaputra Rivers that have a source >5000 m above sea level, whereas that of the Godavari River is at ~900 m. The lower flow velocity of the Godavari River likely causes coarser particles to sink rather than to be transported in the lower water mass as happens in the Amazon and Ganges-Brahmaputra Rivers.*

**Changes made: none.**

Ln 741: Refer to correct figure here.

*Reply: We will correct this.*

**Changes made: we have done this.**